

# 1 Marginal cost curves for water footprint reduction in irrigated agriculture: guiding a
# 2 cost-effective reduction of crop water consumption to a benchmark or permit level

Abebe D. Chukalla[1], Maarten S. Krol[1] and Arjen Y. Hoekstra[1,2]
[1] Twente Water Centre, University of Twente, Enschede, The Netherlands
[2] Institute of Water Policy, Lee Kuan Yew School of Public Policy, National University of Singapore, Singapore
Correspondence to: Abebe D. Chukalla (a.d.chukalla@utwente.nl)

## 9 Abstract

Reducing the water footprint (WF) of the process of growing irrigated crop is an indispensable element in water
management, particularly in water-scarce areas. To achieve this, information on marginal cost curves (MCCs) that
rank management packages according to their cost-effectiveness to reduce the WF need to support the decision
making. MCCs enable the estimation of the cost associated with a certain WF reduction target, e.g. towards a
given WF permit (expressed in $m^3$ per hectare per season) or to a certain WF benchmark (expressed in $m^3$ per
tonne of crop). This paper aims to develop MCCs for WF reduction for a range of selected cases. The soil-water-
balance and crop-growth model, AquaCrop, is used to estimate the effect on evapotranspiration and crop yield
and thus WF of crop production due to different management packages. A management package is defined as
specific combination of management practices: irrigation technique (furrow, sprinkler, drip or subsurface drip);
irrigation strategy (full or deficit irrigation); and mulching practice (no, organic or synthetic mulching). The annual
average cost for each management package is estimated as the annualised capital cost plus the annual costs of
maintenance and operations (i.e. costs of water, energy, and labour). Different cases is considered, including:
three crops (maize, tomato and potato); four types of environment; three hydrologic years (wet, normal and dry
years) and three soil types (loam, silty clay loam and sandy loam). For each crop, alternative WF reduction
pathways were developed, after which the most cost-effective pathway was selected to develop the MCC for WF
reduction. When aiming at WF reduction one can best improve the irrigation strategy first, next the mulching
practice and finally the irrigation technique. Moving from a full to deficit irrigation strategy is found to be a no-
regret measure: it reduces the WF by reducing water consumption at negligible yield reduction, while reducing
the cost for irrigation water and the associated costs for energy and labour. Next, moving from no to organic
mulching has a high cost-effectiveness, reducing the WF significantly at low cost. Finally, changing from sprinkler
or furrow to drip or sub-surface drip irrigation reduces the WF but at significant cost.

Key words: water abatement cost curve, water saving, irrigation practice, soil water balance, crop growth, crop
modelling



## 1. Introduction

In many places, water use for irrigation is a major factor contributing to water scarcity (Rosegrant et al., 2002;Mekonnen and Hoekstra, 2016), which will be enhanced by increasing demands for food and biofuels (Ercin and Hoekstra, 2014). In many regions, climate change will aggravate water scarcity by affecting the spatial patterns of precipitation and evaporation (Vörösmarty et al., 2000;Fischer et al., 2007). To ensure that WF in a catchment remains within the maximum sustainable level given the water renewal rate in the catchment, Hoekstra (2014) proposes to establish a WF cap per catchment and issue no more WF permits to individual users than fit within the cap. This would urge water users to reduce their WF to a level that is sustainable within the catchment. Additionally, in order to increase water use efficiency, Hoekstra (2014) proposes water footprint benchmarks for specific processes and products as a reference for what is a reasonable level of water consumption per unit of production. This would provide an incentive for water users to reduce their WF per unit of product down to a certain reasonable reference level. The reduction of the WF in irrigated agriculture to the benchmark level relates to improving the physical water use efficiency thus relieving water scarcity (Mekonnen and Hoekstra, 2014;Zhuo et al., 2016;Zwart et al., 2010). WF reduction in irrigated crop production can be achieved through a range of measures, including a change in mulching or irrigation technique or strategy. Chukalla et al. (2015) studied the effectiveness of different combinations of irrigation techniques, irrigation strategy and mulching practice in terms of WF reduction. No research thus far has been carried out regarding the costs of WF reduction. A relevant question though is how much does it cost to reduce the WF of crop production to a certain target such as a WF benchmark for the water consumption per tonne of crop or a WF permit for the water consumption per area.

The current study makes a first effort in response to this question by analysing the cost effectiveness of various measures in irrigated crop production in terms of cost per unit of WF reduction and introducing marginal cost curves (McCraw and Motes) for WF reduction. A MCC for WF reduction is a tool that presents how different measures can be applied subsequently in order to achieve an increasing amount of WF reduction, whereby measures are ordered according to their cost effectiveness (WF reduction achieved per cost unit). Every new measure introduced brings an additional (i.e. marginal) cost and an incremental (marginal) reduction of the WF. MCCs have been extensively applied to the case of carbon footprint reduction in different sectors (industries, agriculture etc.) (MacLeod et al., 2010;Enkvist et al., 2007;Kesicki, 2010;Bockel et al., 2012). In the case of other environmental footprints (Hoekstra and Wiedmann, 2014), like the ecological and water footprint, the MCC application is just starting (Khan et al., 2009;Tata-Group, 2013). In the area of water, Addams et al. (2009) has applied MCCs to analyse the cost of reducing water withdrawals for irrigated agriculture, but they didn't analyse the cost in relation WF reduction. Introducing the MCC for WF reduction given the increasing water scarcity and need to enhance water use sustainability and efficiency is imperative.

The objective of this study is to develop alternative WF reduction pathways and the MCC for WF reduction in irrigated crop production. We do so for a number of crops and environments. We apply the AquaCrop model, a





soil-water-balance and crop-growth model that can be used to estimate the WF of crop production under
different management practices, linked with a cost model that calculates annual costs related to different
management practices, to systematically assess both WF and costs of twenty management packages. Based on
the outcomes we construct WF reduction pathways and marginal cost curves. Finally, we illustrate the application
of the MCC for WF reduction with a selected case with a certain WF reduction target given a situation where the
actual WF needs to be reduced given a cut in the WF permit.

2.  **Method and data**

*2.1  Research set-up*

We consider the production of three crops (maize, tomato and potato) under four environments (humid, sub-
humid, semi-arid and arid), three hydrologic years (wet, normal and dry year) and three soil types (loam, silty clay
loam and sandy loam). We distinguish twenty management packages, whereby each management package is
defined as specific combination of management practices: irrigation technique (furrow, sprinkler, drip or
subsurface drip); irrigation strategy (full or deficit irrigation); and mulching practice (no, organic or synthetic
mulching).

We develop the marginal cost curves (MCCs) for WF reduction in irrigated crop production in four steps (Figure
1). First, we calculate the WF of growing a crop under the different environmental conditions and management
packages using the AquaCrop model (Raes et al., 2013). Second, the total annual average cost for the
management packages were calculated. Third, we constructed plausible WF reduction pathways starting from
different initial situations. A WF reduction pathway shows a sequence of complementary measures, stepwise
moving from an initial management package to management packages with lower WFs. Finally, MCCs for WF
reduction were deduced based on reduction potential and cost effectiveness of the individual steps. This
approach does not aim to represent a cost-benefit analysis from an agro-economic perspective. Reduced costs
through water savings are included, but monetary benefits to the farmer through increased yield or product
quality are not included. In this way, the approach fully focusses on costs to save water. Yield increases do have
a direct impact on final results by reducing the WF per unit of product.





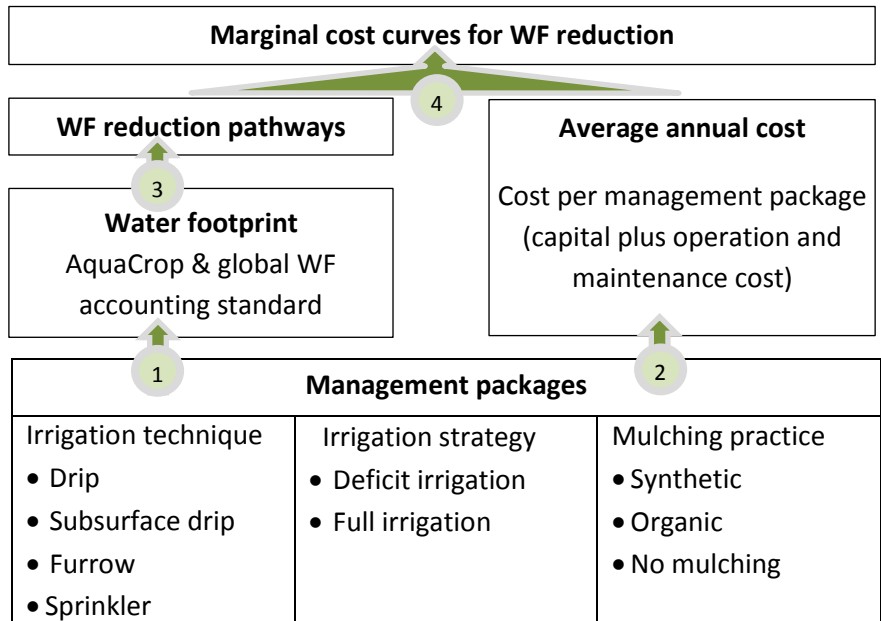

**Figure 1.** Flow chart for developing marginal cost curves for crop production

*2.2 Management packages*

Each management package is a combination of a specific irrigation technique, irrigation strategy and mulching practice. We consider four irrigation techniques, two irrigation strategies and three mulching practices. From the 24 possible combinations, we exclude four unlikely combinations, namely the combinations of furrow and sprinkler techniques with synthetic mulching (with either full or deficit irrigation), leaving 20 management packages considered in this study.

The four irrigation techniques differ considerably in the wetted area generated by irrigation (Ali, 2011). In the analysis, default values from the AquaCrop model are taken for the wetted area for each irrigation, as recommended by (Raes et al., 2013). For furrow irrigation, an 80% wetting percentage is assumed to be representative for a narrow bed furrow, from the indicative range of 60% to 100%. For sprinkler, drip and subsurface drip irrigation techniques, wetted areas by irrigation of 100%, 30% and 0%, respectively, are used.

Two irrigation strategies are analysed: full and deficit irrigation. Irrigation requires two principal decisions of scheduling: the volume of water to be irrigated and timing of irrigation. Full irrigation is an irrigation strategy in which the full evaporative demand is met; this strategy aims at maximizing yield. Its irrigation schedule is





simulated through automatic generation of the required irrigation to avoid any water stress. The irrigation
schedule in the no water stress condition is crop-dependent: the soil moisture is refilled to field capacity (FC)
when 20%, 36% and 30% of readily available water (RAW) of the soil is depleted for maize, potato and tomato
respectively (FAO, 2012). This scheduling results in a high irrigation frequency, which is impractical in the case of
furrow and sprinkler irrigation. To circumvent such unrealistic simulation for the case of furrow and sprinkler
irrigation, the simulated irrigation depths are aggregated in such a way that a time gap of a week is maintained
between two irrigation events.
Deficit irrigation (DI) is the application of water below the evapotranspiration requirements (Fereres and Soriano,
2007) by limiting water applications particularly during less drought-sensitive growth stages (English, 1990). The
deficit strategy is established by reducing the irrigation supply below the full irrigation requirement. We
extensively tested various deficit irrigation strategies that fall under two broad categories: (1) regulated deficit
irrigation, where a non-uniform water deficit level is applied during the different phenological stages; and
sustained deficit irrigation, where the water deficit is managed to be uniform during the whole crop cycle. In the
analysis of simulations, the specific deficit strategy that is optimal according to the model experiments and for
yield reduction not exceeding 2% is used. AquaCrop simulates water stress responses triggered by soil moisture
depletion using three thresholds for a restraint on canopy expansion, stomatal closure and senescence
acceleration (Steduto et al., 2009b).
Mulching is the process of covering the soil surface around a plant to create good-natured conditions for its
growth (Lamont et al., 1993;Lamont, 2005). Mulching has various purposes: reduce soil evaporation, control weed
incidence and its associated water transpiration, reduce soil compaction, enhance nutrient management and
incorporate additional nutrients (McCraw and Motes, 1991;Shaxson and Barber, 2003;Mulumba and Lal, 2008).
The AquaCrop model simulates the effect of mulching on evaporation and represents effects of soil organic
matter through soil hydraulic properties influencing the soil water balance. Soil evaporation under mulching
practices is simulated by scaling the evaporation with a factor that is described by two variables (Raes et al.,
2013): the fraction of soil surface covered by mulch (from 0 to 100 %); and a parameter representing mulch
material ($f_m$). The correction factor for the effect of mulching (CF) on evaporation is calculated as:
$$CF = (1 - f_m \frac{\% \text{ covere by mulch}}{100})$$  (1)
We assume a mulching factor $f_m$ of 1.0 for synthetic mulching, 0.5 for organic mulching and zero for no mulching
as suggested by Raes et al. (2013). Further we take a mulch cover of 100 % for organic and 80 % for synthetic
materials, again as suggested in the AquaCrop reference manual (Raes et al., 2013).
*2.3 Calculation of water footprint per management package*






The water footprint (WF) of crop production is a volumetric measure of fresh water use for growing a crop,
distinguishing between the green WF (consumption of rainwater), blue WF (consumption of irrigation water or
consumption of soil moisture from capillary rise) and the grey WF (water pollution) (Hoekstra et al., 2011). The
green and blue WF, which are the focus in the current study, are together called the consumptive WF. To allow
for a comprehensive and systematic assessment of consumptive WF, this study employs the AquaCrop model to
estimate green and blue evapotranspiration (ET) and crop yield (Y) to calculate blue and green WF of crop
production.

We use the plug-in version of AquaCrop 4.1 (Steduto et al., 2009a;Raes et al., 2011) and determine the crop
growing period based in growing degree days. AquaCrop model simulates the soil water balance in the root zone
with a daily time step over the crop growing period (Raes et al., 2012). The fluxes into and from the root zone are
runoff, infiltration, evapotranspiration, drainage and capillary rise. The green and blue fractions in total ET are
calculated based on the green to blue water ratio in the soil moisture, which in turn is kept track of over time by
accounting for how much green and blue water enter the soil moisture, following the accounting procedure as
reported in (Chukalla et al., 2015).

AquaCrop simulates actual ET and biomass growth based on the type of crop grown (with specific crop
parameters), the soil type, climate data such as precipitation and reference ET ($ET_o$), and given water and field
management practices. We estimate $ET_o$ based on FAO's $ET_o$ calculator that uses the Penman-Monteith equation
(Allen et al., 1998). The model separates daily ET into crop transpiration (productive) and soil evaporation (non-
productive). Evaporation (E) is calculated by multiplying reference ET ($ET_o$) with factors that consider the fraction
of the soil surface not covered by canopy, and water stress. There are two stages of determining evaporation
(Ritchie, 1972):  the first stage,  the evaporation rate is fully determined by the energy available for soil
evaporation – this happens when the soil surface is soaked by rainfall or irrigation; and the second stage also
called the falling rate stage, the evaporation is determined by the available energy and on the hydraulic properties
of the soil. AquaCrop model is confirmed to reasonably simulate evaporation, transpiration and thus ET through
field experimental studies for various conditions (Afshar and Neshat, 2013;Saad et al., 2014).

The crop growth engine of AquaCrop estimates the biomass by multiplying water productivity and transpiration
and yield by multiplying biomass with harvest index.  Water productivity is assumed to respond to atmospheric
evaporative demand and atmospheric $CO_2$ concentration (Steduto et al., 2009a).

We express the WF of crop production in two ways. The green and blue WF per unit of land (m³/ha) are calculated
as the green and blue evapotranspiration over the growing period of a crop. The green and blue WF per unit of
production (m³/tonne) are calculated by dividing green or blue evapotranspiration over the growing period of a
crop (m³/ha) by the crop yield (tonne/ha). The crop yield in terms of dry matter per hectare as obtained from the
AquaCrop calculations is translated into a fresh crop yield (the marketable yield) per hectare. The dry matter



fractions of marketable yield for tomato, potato and maize are estimated to be 7%, 25% and 100%, respectively
(Steduto et al., 2012). The variability of green and blue WF are presented by calculating the standard deviation of
the estimated WFs across different environments, hydrologic years and soil types.

*2.4 Estimation of annual cost per management package*

The overall cost of a management package includes initial capital or investment costs (IC), operation costs (OC),
and maintenance costs (MC). Investment costs include costs of installing a new irrigation system and/or buying
plastics for synthetic mulching. Operation cost refer to costs for irrigation water, energy and labour. Maintenance
costs include labour and material costs. Both OC and MC are expressed as annual cost (US$/ha per year).

Figure 2 shows the average annual investment cost of irrigation techniques and their lifespan. The data are
derived from different sources as specified in Appendices A and B. Investment costs that were reported as one-
time instalment costs were converted to equivalent annual costs based on a 5% interest rate and the lifespan of
the techniques. The average annual maintenance cost per irrigation technique – including costs for labour and
material – is assumed to be equivalent to 2% of the annualised investment costs (Kay and Hatcho, 1992).

The average annual investment costs of US$ 1112 per ha for synthetic mulching is based on the sources as
specified in Appendix C. We further assume an average operation and maintenance costs of US$ 140 per ha per
year for synthetic mulching and US$ 200 per ha per year for organic mulching.

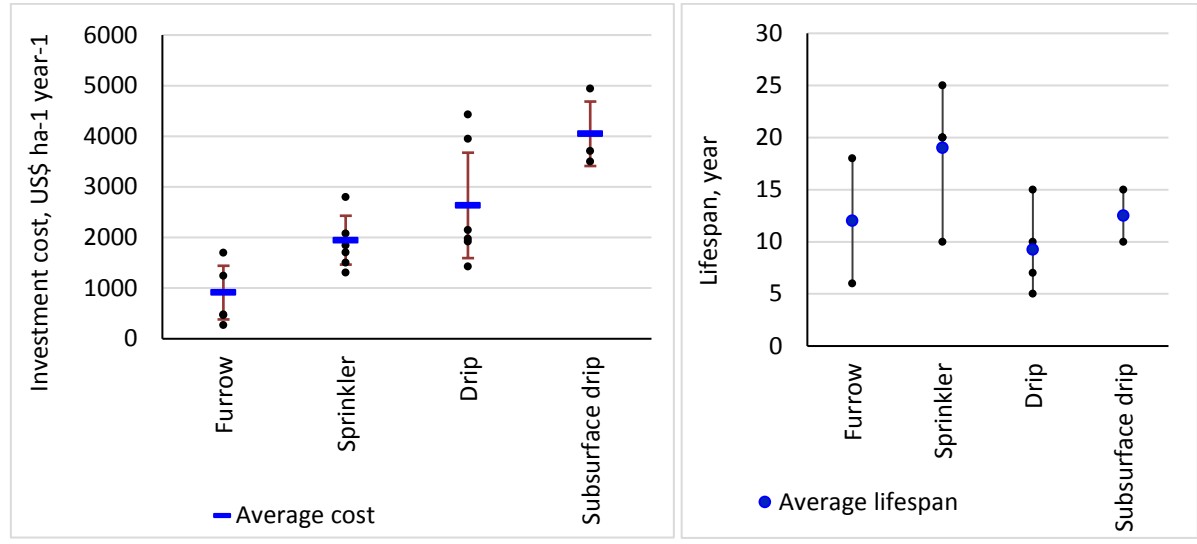

**Figure 2.** Annual investment cost and lifespan for irrigation techniques

The operational cost related to the use of irrigation water is calculated from the amount of irrigation water
applied and an average unit price of water (0.09 ± 0.02 US$ per m³, Appendix E). The amount of irrigation supply





is calculated by dividing the irrigation volume applied at field level simulated by AquaCrop by the application
efficiency (Phocaides, 2000). Application efficiency, the ratio of actually applied to supplied irrigation water, is
different per irrigation technique (Table 1). The operational cost related to energy use for sprinkler, drip and
subsurface drip irrigation is calculated as the total energy demand over the growing season multiplied by the cost
of energy (Appendix F). The total energy demand (kWh) is calculated as follows (Kay and Hatcho, 1992):

$$\text{Seasonal energy demand } = \frac{I \times h}{367\eta} \tag{2}$$

where I is the volume of irrigation water to be pumped in a crop season (in m$^3$), h the pressure head (in m) given
in Table 1.
and $\eta$ the pump efficiency. The pump efficiency can be between 40% and 80% for a pump running at optimum
head and speed and is assumed at 60% here (Kay and Hatcho, 1992).
Energy required to transport surface water to the field or to pump up groundwater is not included in the
estimates.

The operational cost related to labour is calculated as the required labour hours per irrigation event times the
number of irrigation events times the cost of labour per h. The number of irrigation events in the crop growing
period is simulated with AquaCrop. The required labour hours per irrigation event is shown in Table 1 and the
cost of labour per h is given in Appendix D.

**Table 1.** The application efficiency, labour intensity and pressure head required per irrigation technique.

| Irrigation technique | Application efficiency (%) | Labour intensity (h ha$^{-1}$ per irrigation event) | Pressure head (m) |
|---|---|---|---|
| | Sources: Brouwer et al. (1989), Kay and Hatcho (1992), and Phocaides (2000) | Sources:Kay and Hatcho (1992) | Sources: (Reich et al., 2009) and Phocaides (2000) |
| Furrow | 60 | 2.0-4.0 | 0 |
| Sprinkler | 75 | 1.5-3.0 | 25 |
| Drip | 90 | 0.2-0.5 | 13.6 |
| Subsurface drip | 90 | 0.2-0.6 | 13.6 |


Uncertainties in the cost estimations are represented by their standard deviation. The standard deviations in the
investment and maintenance costs and operational costs for water, energy and labour were systematically
combined in calculating the standard deviation for the total cost estimation.



### 2.5 Marginal cost curves for WF reduction

After having calculated the total cost and WF associated with each management package, the MCC for reducing the WF per area or per unit of crop in irrigated agriculture is developed in two steps:

1. Identify alternative WF reduction pathways by arranging plausible progressive sequences of management packages from a baseline management package to a management package with the smallest WF.
2. Select the most cost-effective pathway for a certain baseline and derive from that pathway the MCC for WF reduction.

We consider two baseline management packages: the full irrigation strategy and no mulching practice combined with either furrow or sprinkler irrigation. These two management packages are the most widely deployed types of water and field management (Baldock et al., 2000).

The marginal cost (MC) of a unit WF reduction for a management package 2 compared to a preceding management package 1 along the pathway of decreasing WF is calculated as:

$$\text{MC of a unit WF reduction} = \frac{(TC_2 - TC_1)}{WF_1 - WF_2} \tag{3}$$

where $TC_x$ refers to the total annual cost of management package x and $WF_x$ to the water footprint of management package x.

The MCC shows how subsequent WF reductions can be achieved in the most cost-effective way by moving from the baseline management package to another package, and further to yet another package and so on. It shows both cost and WF reduction achieved with each step. With each step, the marginal cost of WF reduction will increase.

### 2.6 Data

The WFs were calculated for four locations (Israel, Spain Italy and the UK), three hydrological years (wet, normal and dry years) and three soil types (loam, silty clay loam, sandy loam). The input data on climate and soil were collected from four sites: Rothamsted in the UK (52.26° N, 0.64°E; 69m amsl), Bologna in Italy (44.57 °N, 11.53 °E; 19m amsl), Badajoz in Spain (38.88 °N, -6.83 °E; 185m amsl), and Eilat in Israel (29.33 °N, 34.57 °E; 12m above mean sea level). These sites characterise humid, sub-humid, semi-arid, and arid environments respectively. Daily observed climatic data (rainfall, minimum and maximum temperature), were extracted from the European Climate Assessment and Dataset (ECAD) (Klein Tank et al., 2002). Wet, normal and dry years were selected from 20 years daily rainfall data (observed data from the period 1993 to 2012). Daily $ET_o$ for the wet, normal and dry



years were derived using FAO's $ET_o$ calculator (Raes, 2012). Soil texture data, which is extracted with the
resolution of 1×1km$^2$ from European Soil Database (Hannam et al., 2009), is used to identify the soil type based
on the Soil Texture Triangle calculator (Saxton et al., 1986). The physical characteristics of the soils is used taken
from the default parameters in AquaCrop. For crop parameters, by and large we take the default values as
represented in AquaCrop. However, the rooting depth for maize at the Bologna site is restricted to the maximum
of 0.7m to account for the actual local condition of a shallow groundwater table (average 1.5 m). The main
components of the average annual cost per management package have been collected from literature. The costs
for water, labour and energy are averaged over data for Spain, Italy and the UK, i.e. from three of the four
countries studied here. An overview of the costs and their sources are presented in Appendices A to F.
**3. Results**
3.1 Water footprint and cost per management package
Figures 3 and 4 show the WF per area and WF per unit of crop, and the annual average costs corresponding to
twenty management packages.
For each combination of a certain mulching practice and irrigation strategy, the consumptive WF and the blue WF
in particular decrease when we move from sprinkler to furrow to drip and further to subsurface drip irrigation.
Under given irrigation strategy and mulching practice, the WF in m$^3$/ha in case of subsurface drip irrigation is 6.2-
13.3% smaller than in case of sprinkler irrigation. The annual average cost always increases from furrow to
sprinkler and further to drip and subsurface drip irrigation. Under given mulching practice and irrigation strategy,
the cost in case of furrow irrigation is 58-63% smaller than in case of subsurface drip irrigation. The cost of furrow
irrigation is small particularly because of the relatively low investment cost, which is higher for sprinkler and even
higher for drip and subsurface drip irrigation. The operational costs, on the contrary, are higher for sprinkler and
furrow than for drip or subsurface drip irrigation, because of the higher water consumption and thus cost for
sprinkler and furrow. Sprinkler has the highest operational cost because it requires a high pressure head to
distribute the water (thus higher energy cost).
Under given irrigation technique and mulching practice, deficit irrigation (DI) always results in a slightly smaller
WF in m$^3$/ha (in the range of 1.6-5.7%) and lower cost (in the range of 4-14%) as compared to full irrigation (FI).
The decrease in cost is due to the decrease in water and pumping energy. The WF of crop production always
reduces in a stepwise way when going from no mulching to organic mulching and then to synthetic mulching,
while the costs increase along the move. This cost increase relates to the growing material and labour costs when
applying mulching (most with synthetic mulching), but the net cost increase is tempered by the fact that less
water and pumping energy will be required.



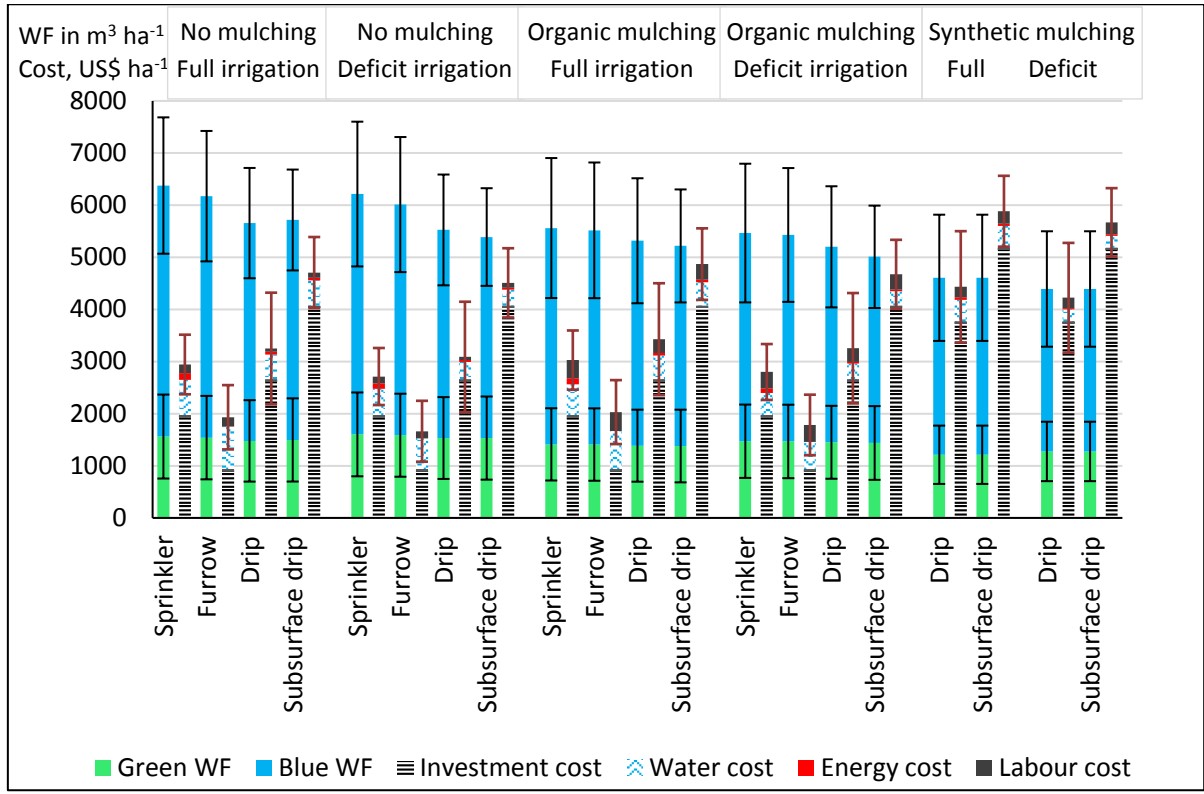

**Figure 3:** Average WF per area (m³ ha⁻¹) for maize production and average annual costs associated with 20 management packages. The whiskers around WF estimates indicate the range of outcomes for the different cases (different environments, hydrologic years and soil types). The whiskers around cost estimates indicate uncertainties in the costs. WF estimates are split up in blue and green components; costs are split up in investment, water, energy and labour costs.



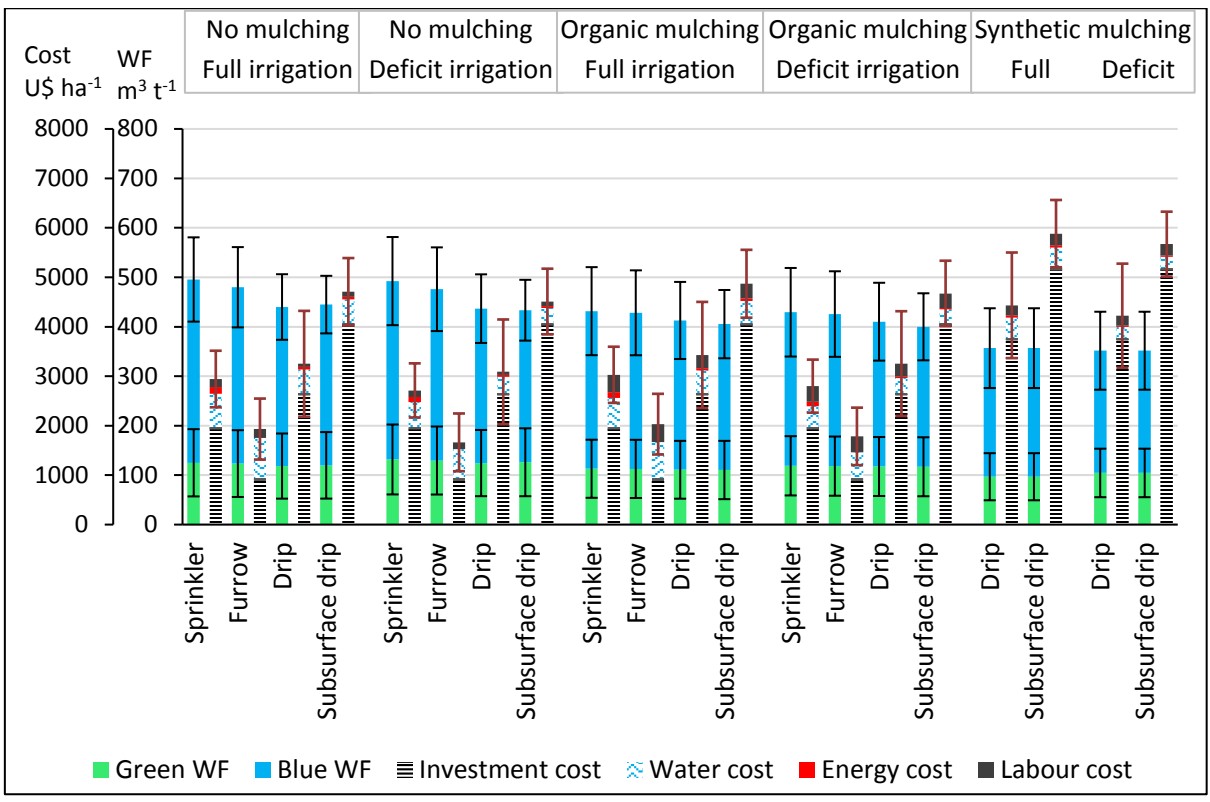

Figure 4: Average WF per product unit (m³ t⁻¹) for maize production and average annual costs associated with 20 management packages. The whiskers around the WF estimates indicate the range of outcomes for the different cases (different environments, hydrologic years and soil types). The whiskers around cost estimates indicate uncertainties in the costs.

Figure 5 shows the scatter plot of the twenty management packages, the abscissa and ordinate of each point representing the average annual cost and average WF, respectively, of a particular management package. In this graph, the blue arrow indicates the direction of decreasing WF and costs. The points or management packages connected by the blue line are jointly called the Pareto optimal front or non-dominated Pareto optimal solutions. Moving from one to another management package on the line will reduce either cost or WF but increase the other, thus implying a trade-off between the two variables. Each management package that is not on the line can be improved in terms of reducing cost or reducing WF or both.



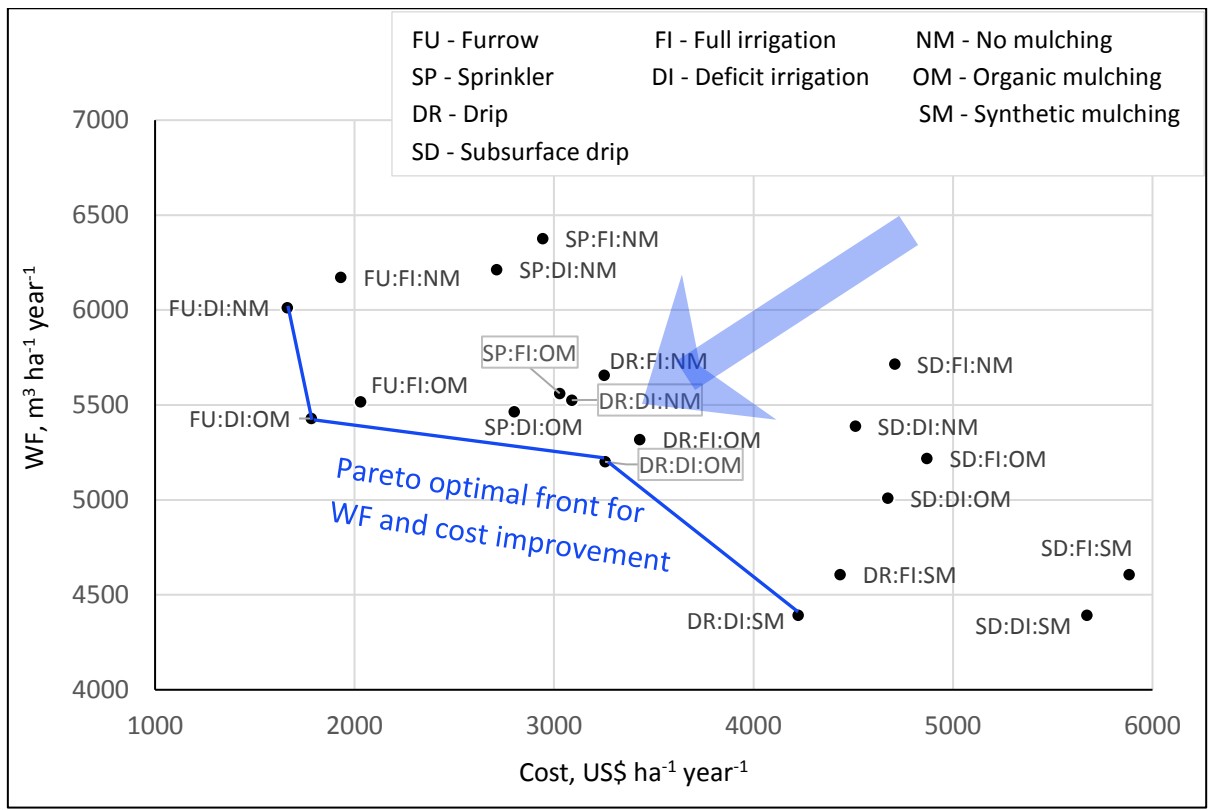

Figure 5: Pareto optimal front for WF and cost reduction in irrigated crop production. The dots represent the annual cost of maize production and the WF per area for twenty management packages. The line connects the Pareto optimal management packages.

## 3.2 Water footprint reduction pathways

In developing a new irrigation scheme or renovating an existing one in a water-scarce area, it would be rational to implement one of the management packages from the Pareto optimal set if the goal is to arrive at a cost-effective minimization of the WF of crop production. In an existing farm, where the management package is not in the Pareto optimal set, there can be alternative pathways towards reducing the WF. This involves a stepwise adoption of complementary measures that eventually leads to a management package in the Pareto optimal set.

Figure 6 shows alternative WF reduction pathways from the two most common baseline management packages: full irrigation and no mulching with either furrow or sprinkler irrigation. The figure shows four WF reduction pathways from the baseline with furrow irrigation and two pathways from the baseline with sprinkler irrigation. In all pathways, the WF of crop production is continually reduced by changing one thing at a time, i.e. either the irrigation technique, the irrigation strategy or the mulching practice. In some cases, a step may be accompanied





by a cost reduction, but in the end most steps imply a cost increase. Logically, all pathways end at a point at the
Pareto optimal front.

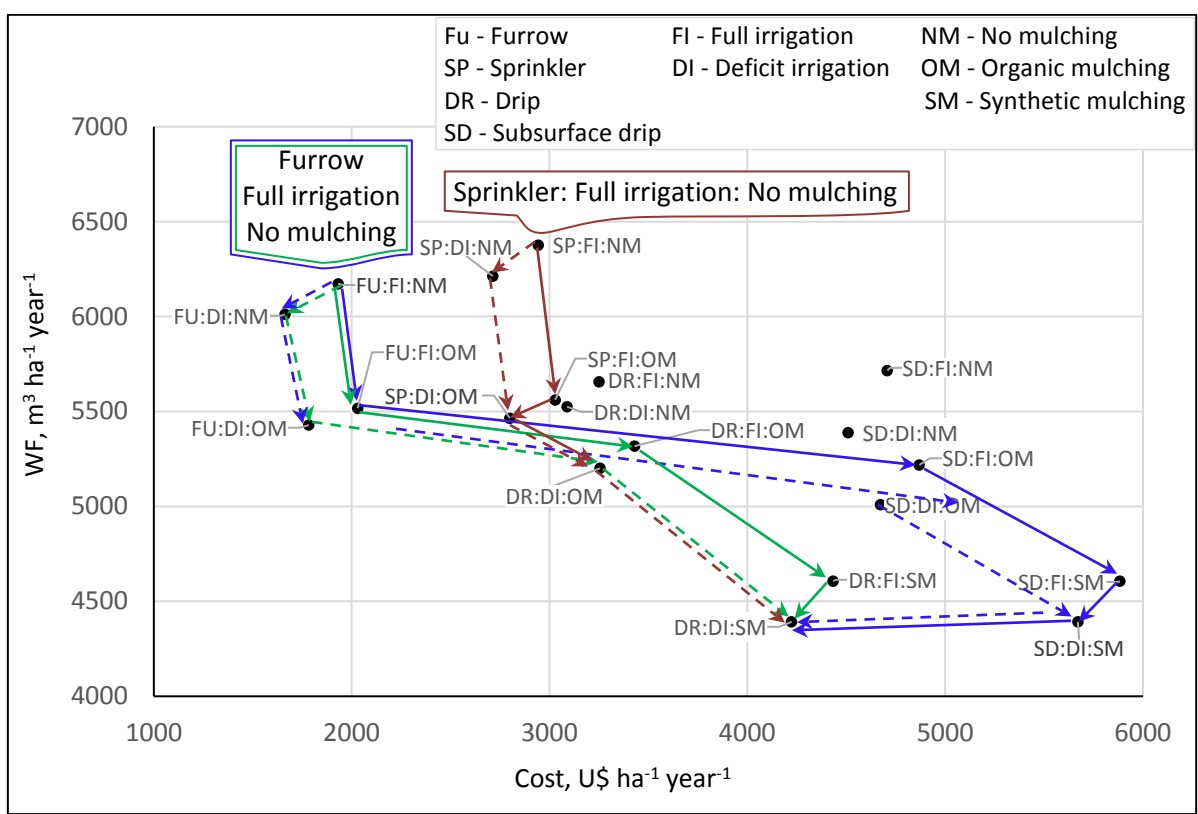

Figure 6: WF reduction pathways for maize from two baseline management packages: full irrigation and no
mulching with either furrow or sprinkler irrigation.

*3.3 Marginal cost curves for WF reduction*

Not all alternative WF reduction pathways from a specific baseline are equally cost effective. In both cases it
makes much sense to move from full to deficit irrigation first, because that reduces the WF and cost at the same
time. Next, it is best to move from no to organic mulching because the cost-effectiveness of this measure is very
high, which can be measured in the graph (Figure 6) as the steep slope (high WF reduction per dollar). Finally, the
most cost-effective measure, in both cases, is to move towards drip irrigation in combination with synthetic
mulching. One could also move to drip irrigation and stay with organic mulching, which is also Pareto optimal;
the cost of this will be less, but the WF reduction will be less as well. However, moving to drip irrigation in
combination with synthetic mulching is more cost-effective (higher WF reduction per dollar) than moving to drip
irrigation while staying with organic mulching.



For both baseline management packages, we have drawn the MCCs in Figures 7 and 8 for the case of maize. The
curves are shown both for reducing the WF per area (Figures 7a and 8a) and the WF per unit of product (Figure
7b and 8b). From these curves, we can read the most cost-effective measures that can subsequently be
implemented. For each step we can read in the graph what is the associated marginal cost and what is the
associated WF reduction. In both cases, the first step goes at a negative cost, i.e. a benefit, while next steps go at
increasing marginal cost. Each step is shown in the form of a bar, with the height and width representing the cost
per unit WF reduction and the WF reduction, respectively. The area under a bar represents the total cost of
implementing the measure.

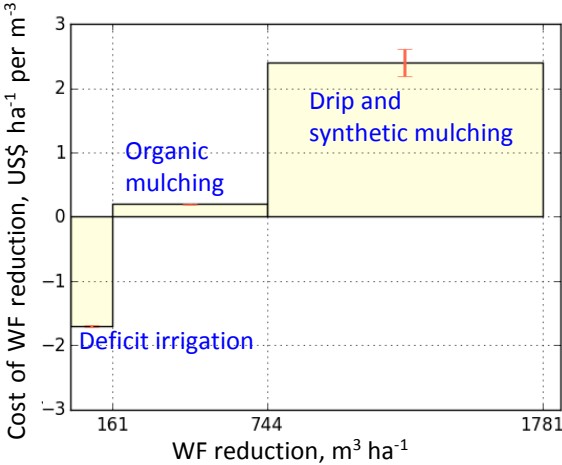
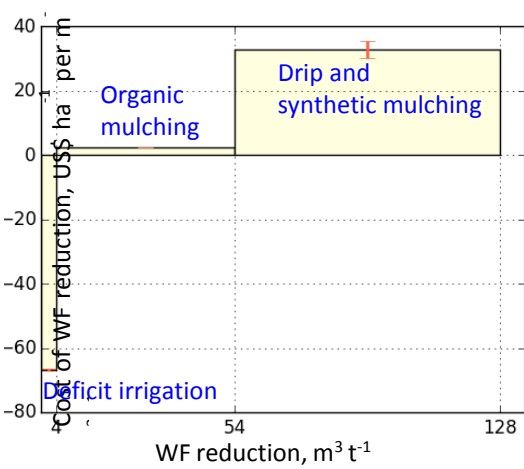


Figure 7: Marginal cost curves for WF reduction for maize for the baseline of furrow technique combined with full
irrigation and no mulching. Left: WF reduction per area. Right: WF reduction per unit of product.

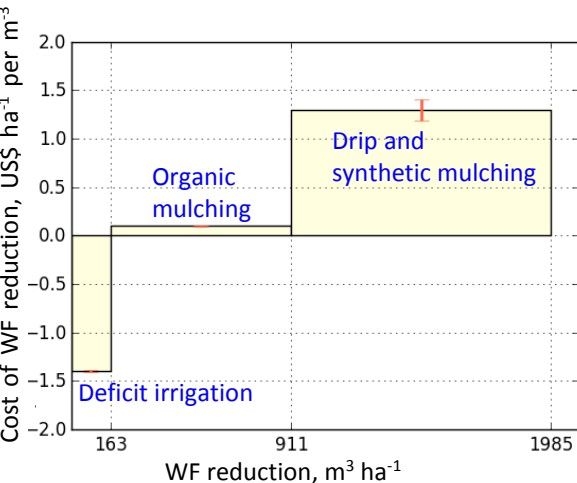
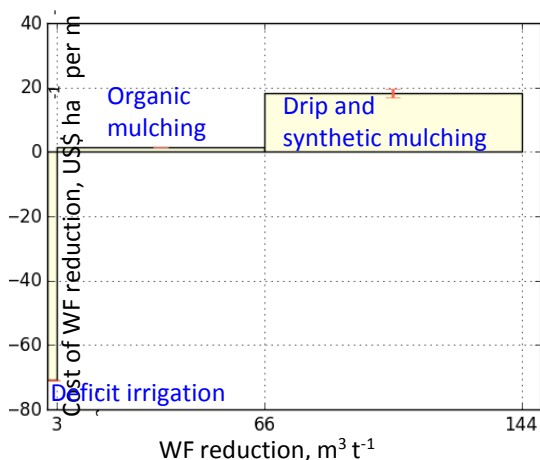


Figure 8: Marginal cost curve for WF reduction of maize for the baseline of sprinkler technique combined with
full irrigation and no mulching. Left: WF reduction per area. Right: WF reduction per unit of product.




For tomato and potato we find similar results as for maize, as shown by the data presented in Appendix G.

*3.4 Application of the marginal cost curve*

In this section, we elaborate a practical application of a MCC for WF reduction, using a selected case with a certain
WF reduction target given a situation where the actual WF needs to be reduced given a cut in the WF permit. The
future introduction of WF permits to water users or WF benchmarks for products in water-scarce areas is likely if
the sustainable development goals (SGDs) are to be met, particularly SDG 6.4, which reads: "by 2030, substantially
increase water-use efficiency across all sectors and ensure sustainable withdrawals and supply of freshwater to
address water scarcity, and substantially reduce the number of people suffering from water scarcity". Here we
will illustrate how a MCC for WF reduction can help in achieving a certain WF reduction goal.

An MCC for WF reduction – ranking measures according to their cost-effectiveness in reducing WF – can be used
to estimate what measures can best be taken and what is the associated total cost to achieve a certain WF
reduction target. For farmers, it will not be attractive to go beyond the implementation of those WF reduction
measures that reduce cost as well, but from a catchment perspective further WF reduction may be required. An
MCC will show the societal cost associated with a certain WF reduction goal. Governments, food companies and
investors can make use of this information to develop incentive schemes for farmers and/or investment plans to
implement the most-cost effective measures in order to achieve a certain WF reduction in a catchment or at a
given farm.

In a hypothetical example, in the river basin the WF exceeds the maximum sustainable level. Agriculture in the
basin consists of irrigated maize production with a current consumptive WF on the farms of 6380 $m^3$ $ha^{-1}$. The
farms apply sprinkler and full irrigation and no mulching. In order to reduce water consumption in the basin to a
sustainable level, the river basin authority proposes various measures including a regulation that prohibits land
expansion for crop production and the introduction of a WF permit to the maize farmers that allows them to use
no more than 5200 $m^3$ $ha^{-1}$. This means they have to reduce the WF of maize production by 1180 $m^3$ $ha^{-1}$. Figure
9 shows how the MCC for WF reduction can help in this hypothetical example to identify what measures can best
be taken to reduce the WF by the required amount and what costs will be involved.

As shown in the figure, we best implement deficit irrigation first (providing a total benefit of 231 USD $ha^{-1}$),
followed by organic mulching (with a total cost of 87 USD $ha^{-1}$). The third and last step to finally achieve the
required WF reduction can be to implement drip irrigation combined with synthetic mulching on 25% of the maize
fields (at a total cost of 356 USD $ha^{-1}$). The other 75% is then still with sprinkler and organic mulching, but the
combined result is meeting the target. Alternatively, because in this particular case the cost-effectiveness of
moving to drip irrigation with organic mulching is close to the cost-effectiveness of moving to drip irrigation with
synthetic mulching, one could move in the third step on 100% of the fields to drip irrigation with organic mulching,




which would result in a WF reduction of 1176 m³ ha⁻¹. In order to meet the full target, a small percentage of the
total fields would need to implement synthetic mulching in addition.

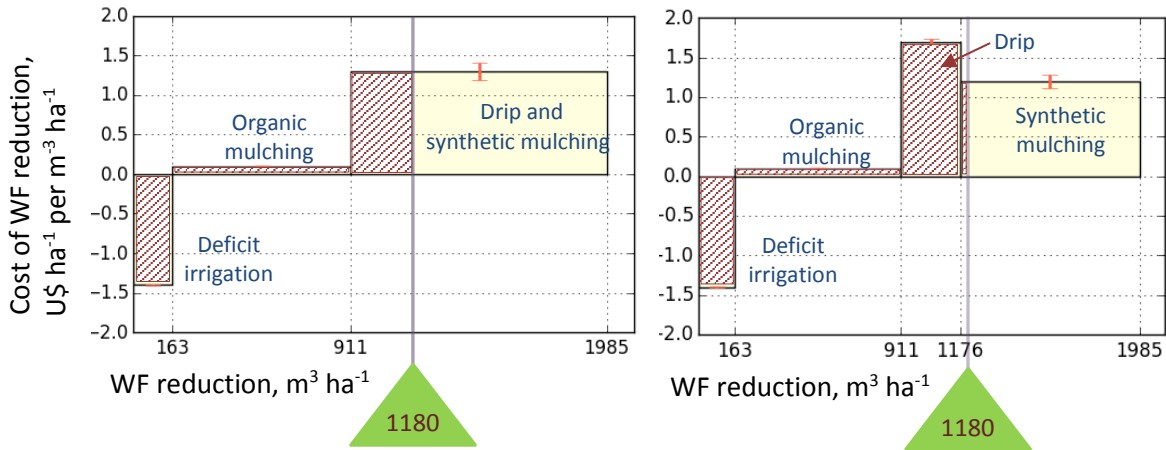

Figure 9: Application of the MCC in an example where the WF of maize production needs to be reduced. The
baseline is sprinkler, full irrigation and no mulching with a WF of 6380 m³ ha⁻¹. This needs to be reduced by 1180
m³ ha⁻¹ in order to meet a given the local WF permit. Left: in the third step, drip irrigation combined with synthetic
mulching is implemented on 25% of the area. Right: in the third step, drip irrigation (maintaining organic
mulching) is implemented on 100% of the area, while in a fourth step synthetic mulching is implemented on 0.5%
of the area.

**4. Conclusion**

In this study, we have developed a method to obtain marginal cost curves for WF reduction in crop production.
The method is innovative by employing a model that combines soil water balance accounting and a crop growth
model and assessing costs and WF reduction for all combinations of irrigation techniques, irrigation strategies
and mulching practices. This is a model-based approach to constructing MCCs, which has the advantage over an
expert-based approach by considering the combined effects of different measures and thus accounting for non-
linearity in the system (i.e. the effect of two measures combined doesn't necessarily equal the sum of the effects
of the separate measures). While this approach has been used in the field of constructing MCCs for carbon
footprint reduction (Kesicki, 2010), this has never been done before for the case of water footprint reduction.

Developing the MCC for WF reduction for three specific irrigated crops, we found that when aiming at WF
reduction one can best improve the irrigation strategy first, next the mulching practice and finally the irrigation
technique. Moving from a full to deficit irrigation strategy is found to be a no-regret measure: it reduces the WF
by reducing water consumption at negligible yield reduction, while reducing the cost for irrigation water and the



associated costs for energy and labour. Next, moving from no to organic mulching has a high cost-effectiveness, reducing the WF significantly at low cost. Finally, changing from sprinkler or furrow to drip or sub-surface drip irrigation reduces the WF but at significant cost.

**Acknowledgments**

This research was conducted as part of the FIGARO, a project funded by the European Commission- as part of the Seventh Framework Program (EU-FP7). The authors thank all the consortium partners in the FIGARO project. The present work was developed within the framework of the Panta Rhei Research Initiative of the International Association of Hydrological Sciences (IAHS

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

**Appendices**

Appendix A: Estimates of the investment cost of irrigation techniques (US$ ha$^{-1}$ year$^{-1}$)

| No | Irrigation techniques | Furrow | Sprinkler | Drip | Subsurface drip |
|----|----------------------|--------|-----------|------|-----------------|
| 1 | Cost | 467 - 1312 | 1844 - 2399 | 1429 - 2594 | |
| | Remark | The techniques are named as surface pumped, sprinkler and localized pumped. The database focuses on the developing regions of the world for the year 2000. | | | |
| | Source | FAO (2016) | | | |
| 2 | Cost | 1700 | 2800 | 3950 | |
| | Remark | Average prices in Europe in 1997. The irrigation technologies are named as improved surface, sprinkler and micro irrigation | | | |
| | Source | Phocaides (2000) | | | |
| 3 | Cost | 1242 | 2080 | 4429 | |





|   |        |                                                                                                                                              |      |      |             |
|---|--------|----------------------------------------------------------------------------------------------------------------------------------------------|------|------|-------------|
|   | Remark | The type of sprinkler is hand moved                                                                                                           |      |      |             |
|   | Source | Custodio and Gurguí (1989)                                                                                                                    |      |      |             |
| 4 | Cost   | 291                                                                                                                                           | 1500 | 1918 | 3500        |
|   | Remark | The one-time investment cost is annualized based on the average life span of the techniques and an interest rate of 5%                         |      |      |             |
|   | Source | Williams and Izaurralde (2006)                                                                                                                |      |      |             |
| 5 | Cost   |                                                                                                                                              |      |      | 3707 - 4942 |
|   | Source | Reich et al. (2009)                                                                                                                           |      |      |             |
| 6 | Cost   |                                                                                                                                              | 1305 | 1976 |             |
|   | Source | Zou et al. (2013)                                                                                                                             |      |      |             |
| 7 | Cost   | 271                                                                                                                                           | 1706 | 2147 |             |
|   | Remark | For a case in China for the year 2000. The irrigation techniques are named as improved surface, sprinkler and micro irrigation                |      |      |             |
|   | Source | Mateo-Sagasta et al. (2013)                                                                                                                   |      |      |             |


Appendix B: Estimates of the lifespan of irrigation techniques from various sources

| Irrigation techniques | Lifespan (years)              |                        |    |                                       |                    |    |                     |
|-----------------------|-------------------------------|------------------------|----|---------------------------------------|--------------------|----|---------------------|
| Source                | Oosthuizen et al. (2005)      | Reich et al. (2009)    |    | Williams and Izaurralde (2006)        | Zou et al. (2013)  |    | Average lifespan    |
| Furrow                | 6                             |                        |    | 18                                    |                    |    | 12                  |
| Sprinkler             |                               | 20                     | 25 | 20                                    | 10                 | 20 | 19                  |
| Drip                  | 7                             |                        |    | 10                                    | 5                  | 15 | 9.25                |
| Subsurface drip       |                               | 10                     | 15 |                                       |                    |    | 12.5                |


Appendix C: Estimates for the cost of mulching (US$ ha$^{-1}$ year$^{-1}$)

| Mulching                                      | Average annual investment cost | Operation and maintenance cost | Sources                          |
|-----------------------------------------------|--------------------------------|--------------------------------|----------------------------------|
| Plastic mulching                              | 1227                           |                                | Lamont et al. (1993)             |
|                                               | 875 to 1750                    |                                | Shrefler and Brandenberger (2014)|
|                                               | 585                            | 140                            | Jensen and Malter (1995)         |
| Average cost for plastic mulching cost ± SD   | 1112 ± 434                     | 140                            |                                  |
| Average cost for organic mulching             |                                | 200 ± 100                      | Klonsky (2012)                   |





Appendix D: Labour cost per hour, in European agriculture for selected countries

| Country | Labour cost | Source |
|---|---|---|
| Italy (Euro h$^{-1}$) | 6.87 | |
| Spain (Euro h$^{-1}$) | 4 | |
| UK (Euro h$^{-1}$) | 8.6 | Agri-Info.Eu (2016) |
| Average (Euro h$^{-1}$) | 6.5 | |
| Average ± SD (US\$ h$^{-1}$) | 7.2 ± 2.3 | |


Appendix E: Cost of water

| Country | Water price | Source |
|---|---|---|
| UK (Euro m$^{-3}$) | 0.06 | Lallana and Marcuello (2016) |
| Spain (Euro m$^{-3}$) | 0.07 | Gómez‐Limón and Riesgo (2004) |
| Italy (Euro m$^{-3}$) | 0.1 | Garrido and Calatrava (2010) |
| Average (Euro m$^{-3}$) | 0.08 | |
| Average ± SD (US\$ m$^{-3}$) | 0.09 ± 0.02 | |


Appendix F: Cost of energy, Eurostat (2016)

| Country | Year | | | Average |
|---|---|---|---|---|
| | 2012 | 2013 | 2014 | |
| Italy | 0.178 | 0.172 | 0.174 | 0.17 |
| Spain | 0.12 | 0.12 | 0.117 | 0.12 |
| UK | 0.119 | 0.12 | 0.134 | 0.12 |
| Average (Euro / kWh) | | | | 0.14 |
| Average ± SD (US\$ / kWh) | | | | 0.15 ± 0.03 |




Appendix G: Summary of marginal cost and WF reduction per subsequent measure in the marginal cost curves
for WF reduction in maize, tomato and potato production
Table G-I: Marginal cost and WF reduction per subsequent measure in the MCC for WF reduction in maize
production for the baseline of furrow technique combined with full irrigation and no mulching.

| Measures | Marginal cost | | WF reduction | | Total cost |
|---|---|---|---|---|---|
| | US$ ha$^{-1}$ per m$^3$ ha$^{-1}$ | US$ ha$^{-1}$ per m$^3$ t$^{-1}$ | m$^3$ ha$^{-1}$ | m$^3$ t$^{-1}$ | US$ ha$^{-1}$ |
| Deficit irrigation | -1.7 | -66.7 | 161 | 4 | -269 |
| Organic mulching | 0.2 | 2.4 | 583 | 50 | 120 |
| Drip and synthetic mulching | 2.4 | 32.9 | 1037 | 74 | 2441 |

Table G-II: Marginal cost and WF reduction per subsequent measure in the MCC for WF reduction in maize
production for the baseline of sprinkler technique combined with full irrigation and no mulching.

| Measures | Marginal cost | | WF reduction | | Total cost |
|---|---|---|---|---|---|
| | US$ ha$^{-1}$ per m$^3$ ha$^{-1}$ | US$ ha$^{-1}$ per m$^3$ t$^{-1}$ | m$^3$ ha$^{-1}$ | m$^3$ t$^{-1}$ | US$ ha$^{-1}$ |
| Deficit irrigation | -1.4 | -70.9 | 163 | 3 | -231 |
| Organic mulching | 0.1 | 1.4 | 748 | 63 | 87 |
| Drip and synthetic mulching | 1.3 | 18.3 | 1073 | 78 | 1424 |

Table G-III: Marginal cost and WF reduction per subsequent measure in the MCC for WF reduction in tomato
production for the baseline of furrow, full irrigation and no mulching.

| Measures | Marginal cost | | WF reduction | | Total cost US$ ha$^{-1}$ |
|---|---|---|---|---|---|
| | US$ ha$^{-1}$ per m$^{-3}$ ha$^{-1}$ | US$ ha$^{-1}$ per m$^{-3}$ t$^{-1}$ | m$^{-3}$ ha$^{-1}$ | m$^{-3}$ t$^{-1}$ | |
| Deficit irrigation | -0.4 | -256.1 | 752 | 1 | -331 |
| Organic mulching | 0.2 | 16.0 | 750 | 8 | 122 |
| Drip and synthetic mulching | 2.3 | 270.2 | 1094 | 9 | 2487 |

Table G-IV: Marginal cost and WF reduction per subsequent measure in the MCC for WF reduction in tomato
production for the baseline of sprinkler technique combined with full irrigation and no mulching.

| Measures | Marginal cost | | WF reduction | | Total cost US$ ha$^{-1}$ |
|---|---|---|---|---|---|
| | US$ ha$^{-1}$ per m$^{-3}$ ha$^{-1}$ | US$ ha$^{-1}$ per m$^{-3}$ t$^{-1}$ | m$^{-3}$ ha$^{-1}$ | m$^{-3}$ t$^{-1}$ | |



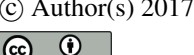

| Deficit irrigation | -0.4 | -275.5 | 840 | 1 | -323 |
|---|---|---|---|---|---|
| Organic mulching | 0.1 | 7.4 | 1045 | 10 | 73 |
| Drip irrigation | 1.4 | 143.2 | 1086 | 4 | 502 |
| Synthetic mulching | | 153.7 | | 6 | 983 |


Table G-V: Marginal cost and WF reduction per subsequent measure in the MCC for WF reduction in potato
production for the baseline of furrow, full irrigation and no mulching.

| Measures | Marginal cost | | WF reduction | | Total |
|---|---|---|---|---|---|
| | US$ ha$^{-1}$ per m$^3$ ha$^{-1}$ | US$ ha$^{-1}$ per m$^3$ t$^{-1}$ | m$^3$ ha$^{-1}$ | m$^3$ t$^{-1}$ | cost US$ ha$^{-1}$ |
| Deficit irrigation | -0.8 | -40.8 | 191 | 4 | -157 |
| Organic mulching | 0.5 | 11.9 | 323 | 12 | 146 |
| Drip and synthetic mulching | 6.2 | 174.8 | 429 | 15 | 2660 |


Table G-VI: Marginal cost and WF reduction per subsequent measure in the MCC for WF reduction in potato
production for the baseline of sprinkler, full irrigation and no mulching.

| Measures | Marginal cost | | WF reduction | | Total cost |
|---|---|---|---|---|---|
| | US$ ha$^{-1}$ per m$^3$ ha$^{-1}$ | US$ ha$^{-1}$ per m$^3$ t$^{-1}$ | m$^3$ ha$^{-1}$ | m$^3$ t$^{-1}$ | US$ ha$^{-1}$ |
| Deficit irrigation | -0.7 | -33.1 | 228 | 5 | -157 |
| Organic mulching | 0.4 | 9.6 | 403 | 15 | 147 |
| Drip and synthetic mulching | 3.5 | 101.6 | 458 | 16 | 1623 |
