# Peer review of "Marginal cost curves for water footprint reduction in irrigated agriculture: guiding a cost-effective reduction of crop water consumption to a permit or benchmark level"

_Hydrology and Earth System Sciences, 2017_

## Referee Comment (RC1) · Anonymous Referee #1 · 3 Mar 2017

The manuscript presents the first attempt to derive MCC for WF reduction. This way the authors add the cost dimension to the water footprint assessment that has not been done before. This is a very timely study and could be interesting for wider audience. The paper needs further revision before it got accepted. The introduction and the discussion section need to further expanded. Please look my detailed comments below:

**The introduction section is very limited. The authors argue that the MCC has not been used in the WF study (Line 474-475) but they fail to carry out a a good literature**

[Figure]

review of the existing literature in the MCC in irrigation water and energy use in irrigated agriculture. I suggest to include some more literature review on the MCC analyses in general and the application of the MCC in irrigation water in particular. There are a number of studies that have been carried out to assess the MC of irrigation water eg. Gonzalez-Alvarez et al. (2006); Samarawickrema and Kulshreshtha (1999). This way, you will put your study in perspective.

**line 250: you are leaving out the major component of the irrigation curve. This is especially very relevant in those water scarce regions where water is pumped from deep groundwater or from far away places (Knutson et al., 1977)! I expect this will change the whole analysis of your MCC. This will further brings up what is the water source, how deep is the groundwater, how far is the surface water, what energy is required to pump the water? The question is then if you include the energy required to bring the water to the field, how will your conclusion change? Do you think, the relative cost saving will warrant the relative yield loss?**

**Pareto optimal state that an allocation is optimal if an action makes someone better off and putting no one worse off. Its weakness is that it doesn't clearly tell which of the Pareto optimal outcomes is best. Besides, it doesn't require equitable use of the water. If that is the case, won't you think it is against one of the pillar of water management "Equitable share" suggested by Hoekstra (2013). Please clearly define the concept clearly and comment on it usefulness to the current study. You might think of using other term.**

**Even in irrigated fields, the contribution of the rainwater (green water) is very significant. To measure the contribution of irrigation to the water use efficiency (water productivity), Bos (1980) suggest the following equations (Howell, 2001):**

$WUEi = (Yi – Yr)/ I$

$WUEet = (Yi – Yr)/ (ETi – ETr)$

where WUEi and WUEet are the contribution of irrigation water to the water use efficiency (WUE) in terms of applied irrigation and actual evapotranspiration, respectively. Yi and Yr, the crop yield under irrigated and rainfed condition, respectively; ETi and ETr, the actual evapotranspiration from irrigated crops and rainfed crops, respectively.

You can define your WF as inverse of the above equations and test it if provides a better insight. At least provide a good argument for choosing to use the WF as in Eqn (3).

**The manuscript could benefit by further discussion of the result, the limitations and recommendations for future improvement or further development and application of the MCC in the WF assessment.**

Minor comments:

**Line 59 McCraw and Motes missing year**

**line 68: add "to" to read "... relation to WF reduction..."**

**line 69; add "the" to read "... the need to enhance ..."**

**line 303: please insert "used is" for "not is used"**

**line 461: please delete "the" from ". . . to meet a given the local WF permit."**

References:

Bos, M.G.: Irrigation efficiencies at crop production level, International Commission on Irrigation and Drainage, Bulletin Vol.29 No.2 pp.18-25, 1980.

Gonzalez-Alvarez, Y., Keeler, A.G. and Mullen, J.D. (2006) Farm-level irrigation and the marginal cost of water use: Evidence from Georgia, Journal of Environmental Management 80: 311–317.

Howell, T. A. 2001. Enhancing Water Use Efficiency in Irrigated Agriculture, Agron. J. 93:281-289. doi:10.2134/agronj2001.932281x

Samarawickrema, A. and Kulshreshtha, S., "Marginal Value of Irrigation Water Use in the South Saskatchewan River Basin, Canada" (2009). Great Plains Research: A Journal of Natural and Social Sciences. Paper 999. http://digitalcommons.unl.edu/greatplainsresearch/999

———————————————

---

## Author Comment (AC1) · 8 Mar 2017

**Reply to Anonymous Referee #1**
We thank Referee #1 for the comments; below we give the reply to the comments.

**Comment**
The manuscript presents the first attempt to derive MCC for WF reduction. This way the authors add the cost dimension to the water footprint assessment that has not been done before. This is a very timely study and could be interesting for wider audience. The paper needs further revision before it got accepted. The introduction and the discussion section need to further expanded. Please look my detailed comments below:
**The introduction section is very limited. The authors argue that the MCC has not been used in the WF study (Line 474-475) but they fail to carry out a good literature review of the existing literature in the MCC in irrigation water and energy use in irrigated agriculture. **I suggest to include some more literature review on the MCC analyses in general and the application of the MCC in irrigation water in particular.** There are a number of studies that have been carried out to assess the MC of irrigation water eg. **Gonzalez-Alvarez et al. (2006); Samarawickrema and Kulshreshtha (1999).** This way, you will put your study in perspective.**

**Reply:** We agree to the reviewer's comment on the originality and timeliness of introducing MCCs for WF reduction, and to the comment on the literature context being presented limitedly. The following remarks will be incorporated in the revised version of the paper.

MCC for WF reduction is a tool showing measures that are ordered according to their cost effectiveness (WF reduction achieved per cost unit) to achieve an increasing amount of WF reduction. Every measure comes with an additional (i.e. marginal) cost and contributes an incremental (marginal) reduction of the WF in crop production. The MCC has been applied extensively in carbon footprint reduction, its application in the area of water footprint is just starting in the industry sector (Tata-Group, 2013).

Previous studies in other thematic domains add the cost dimension to e.g. the management of irrigation water. Addams et al. (2009) apply the MCCs for increasing water supply to close the gap between water supply and demand in irrigated agriculture, particularly focussing on the reduction of irrigation water withdrawal. Khan et al. (2009) discuss two possible pathways to reduce the environmental footprints of water and energy inputs in food production: improving water productivity and energy use efficiency. This work however does not explicitly specify the measures and their cost effectiveness, which would inform the unit cost of improving water and energy use efficiency. Gonzalez-Alvarez et al. (2006) and Samarawickrema and Kulshreshtha (2009) consider the cost dimension in analyzing the management of scarce water resources: the first study makes implications about how farmers would respond if the marginal cost of irrigation water is changed, and the second study assesses the marginal value of irrigation

water in the production of alternative crops in order to allocate the water based on the highest marginal value.

**Comment**

**line 250: you are leaving out the major component of the irrigation curve. This is especially very relevant in those water scarce regions where water is pumped from deep groundwater or from faraway places (Knutson et al., 1977)! I expect this will change the whole analysis of your MCC. This will further brings up what is the water source, how deep is the groundwater, how far is the surface water, what energy is required to pump the water? The question is then if you include the energy required to bring the water to the field, how will your conclusion change? Do you think, the relative cost saving will warrant the relative yield loss?**

**Reply:** We agree with the referee's comment that 'the cost of bringing irrigation water from the source to the field is significant, especially in water scarce regions where water is brought from deep groundwater and/or far away'. The cost of bringing irrigation water to a field is affected by different variables, of which the two most important are the volume of irrigation water and the energy cost required to transport a unit volume of irrigation water. The volume of irrigation water required to grow crops varies with the management at field level (irrigation technology and strategy), while the energy cost to bring a unit volume of irrigation water from a particular source can fairly be assumed equal irrespective of the type of management at field level. The total cost to bring irrigation water from the source to the field, the energy cost per unit volume of water multiplied by the total volume of irrigation water, varies with the source of the irrigation water: the cost is high when the source of water is a deep water well and/or far away, and the cost is low or zero if irrigation water flows to a field due to gravitational force or natural pressure, for example from an artesian aquifer or an elevated reservoir.

In the current study, we are interested to compare the cost effectiveness of measures in reducing the WF of growing crops at field scale, and thus we consider WF of growing a crop and cost of a measure at field scale. The energy cost to bring the water from a source to a field and the water consumption while transporting water between the source and the field level are worth to include, these are however case specific and beyond the scope of the current study. Besides, including these costs would not change the conclucions from the study as we will explain. The overall annual cost per measure increases if we add the cost of energy to bring irrigation water from a source to a field to the annual cost of a measure at field level. The cost increase will depend on the measure (see Fig. 1): the cost increase is highest for furrow irrigation, followed by sprinkler and drip or subsurface drip irrigation; furthermore, the cost increase is more with full irrigation than with deficit irrigation; and finally, the cost increase is most with no-mulching followed by organic and synthetic mulching). One can see that the additional cost related to energy for transporting the water to the field decreases in the direction of decreasing WF. Thus, this does not affect the order of measures ranked based on their cost effectiveness in reducing WF.

We did not include the cost of yield losses because our aim is a cost effectiveness analysis: to see what can be done best at least cost to achieve a certain desired WF reduction.

[Figure]

Figure 1: Average WF per area (m³ ha⁻¹) for maize production and average annual costs associated with 20 management packages. The whiskers around WF estimates indicate the range of outcomes for the different cases (different environments, soils and hydrologic years). The whiskers around cost estimates indicate uncertainties in the costs. WF estimates are split up in blue and green components; costs are split up in investment, water, energy and labour costs. The energy cost to bring irrigation water from a source to a field is calculated by multiplying the volume of irrigation water by a cost of energy, assumed 0.2 $ per m³.

**Comment**

\# Pareto optimal state that an allocation is optimal if an action makes someone better off and putting no one worse off. Its weakness is that it doesn't clearly tell which of the Pareto optimal outcomes is best. Besides, **it doesn't require equitable** use of the water. If that is the case, won't you think it is against one of the pillar of water management "Equitable share" suggested by Hoekstra (2013)Please clearly define the concept clearly and comment on its usefulness to the current study. You might think of using other term.

**Reply:**

Cost-effective WF reduction means to reduce WF at least cost. In the scatter plot showing cost and WF reduction for different management packages, the Pareto optimal set consists of management packages whereby moving from one to another management package will reduce either cost or WF but increase the other, thus implying a trade-off between the two variables. In the uncommon case of the existence of

one "best" solution (no trade-off), the Pareto set would consist of one point. Commonly in multi-objective optimization the Pareto optimal involves trade-offs between the different objectives. "Best solutions" may be identified using the MCC when policy goals are specified e.g. requiring a target WF reduction to be achieved, or a budget to be best spent in reducing WFs.

**Comment**

**Even in irrigated fields, the contribution of the rainwater (green water) is very significant. To measure the contribution of irrigation to the water use efficiency (water productivity), Bos (1980) suggest the following equations (Howell, 2001):**

$WUE_i = (Y_i – Y_r)/ I$

$WUE_{et} = (Y_i – Y_r)/ (ET_i – ET_r)$

where $WUE_i$ and $WUE_{et}$ are the contribution of irrigation water to the water use efficiency (WUE) in terms of applied irrigation and actual evapotranspiration, respectively. $Y_i$ and $Y_r$, the crop yield under irrigated and rain-fed condition, respectively; $ET_i$ and $ET_r$, the actual evapotranspiration from irrigated crops and rain-fed crops, respectively.

You can define your WF as inverse of the above equations and test it if provides a better insight. At least provide a good argument for choosing to use the WF as in Eqn (3).

**Reply:**

In assessing the performance of agricultural management in irrigated crop production, a wide variety of indicators may be used, each stressing a different aspect of what is considered good performance. This may include considering the yield gain per unit of irrigation water applied or per unit of additional ET as a result of irrigation (the two indicators suggested by the reviewer). The introduction of the water footprint, as a policy-relevant indicator, can be found in the first paragraph of the introduction (lines 38-56). The goal of the indicator is to relate human consumption of commodities to appropriation of ET. The choice to subsequently use WF in equation (3) is simply because the goal of the paper is to analyse reductions in water footprints in relation to the costs to achieve these reductions. Lines 176-182 explain that the WF analysed includes both green and blue components. Replacing WF by $WUE_i$, $WUE_{et}$ or yet another performance indicator would yield marginal cost curves that would also be useful, serving different goals. A remark in this direction will be added in the conclusion section.

**Comment**

**The manuscript could benefit by further discussion of the result, the limitations and recommendations for future improvement or further development and application of the MCC in the WF assessment.**

**Reply:**

This paper aims to introduce the methodological development of MCC for WF reduction. In addition we give insights in the interpretation of the MCC by giving a synthetic example. We are not claiming the reported specific values for costs and WF to be valid for any case study; we applied the method for a few specific crops, locations and soils only. Data on costs are taken from various literature sources and averaged over three countries. The average cost of water for the case of the three countries (UK, Spain

and Italy) is 0.09 US$ per m$^3$. The cost of water in UK and Spain is lower than the average while the cost of water in Italy is higher than the average cost.

**Minor comments: (all minor comments are incorporated)**
**Line 59 McCraw and Motes missing year**
**line 68: add "to" to read "... relation to WF reduction..."**
**line 69; add "the" to read "... the need to enhance ..."**
**line 303: please insert "used is" for "not is used"**
**line 461: please delete "the" from ". . . to meet a given the local WF permit."**

References:
Abraham, A., and Jain, L.: Evolutionary multiobjective optimization, in: Evolutionary Multiobjective Optimization, Springer, 1-6, 2005.
Addams, L., Boccaletti, G., Kerlin, M., and Stuchtey, M.: Charting our water future: economic frameworks to inform decision-making, McKinsey & Company, New York, 2009.
Bos, M.: Irrigation efficiencies at crop production level, International Commission on Irrigation and Drainage, Bulletin, 29, 18-60, 1980.
Gonzalez-Alvarez, Y., Keeler, A. G., and Mullen, J. D.: Farm-level irrigation and the marginal cost of water use: Evidence from Georgia, Journal of Environmental Management, 80, 311-317, 2006.
Howell, T. A.: Enhancing water use efficiency in irrigated agriculture, Agronomy journal, 93, 281-289, 2001.
Khan, S., Khan, M., Hanjra, M., and Mu, J.: Pathways to reduce the environmental footprints of water and energy inputs in food production, Food policy, 34, 141-149, 2009.
Samarawickrema, A., and Kulshreshtha, S.: Marginal value of irrigation water use in the South Saskatchewan River Basin, Canada, Great Plains Research, 73-88, 2009.
Tata-Group: Tata Industrial Water Footprint Assessment: Results and Learning, 2013.

---

## Referee Comment (RC2) · Anonymous Referee #2 · 13 Mar 2017

The manuscript is very interesting and focuses on a very important topic: WF reduction studied with the application of MCC for analysing the economic side of strategies improvement for water use reduction. The study is well balanced and clearly written, therefore I suggest accepting it after solving few comments.

Specific comments: In the introduction literature is lacking, more details should be given on MCC, on possible studies that tried to perform something similar, and better explaining the advantages and innovation of introducing such an assessment (e.g., pay more attention on lines 67-70). In addition, the references reported are written often together

(line 64: 4 references in the brackets) and explaining their single specific role as reference would be helpful.

Lines 208-213: what are the average yields for the crops? Maize is considered cultivated in Italy only, or also in Spain for example? What about the other crops? With this point in mind, are the Figures 4-5 referred to maize production in one single country or in more? Understanding the country would make possible to connect these results with the values reported in the Appendix.

Line 250-251: not considering energy for transport and pumping is a very important simplification and surely affects the results. Please motivate your choice.

Finally, a discussion paragraph is missing, which would be helpful for better discussing literature, the benefits of this new method applied to WF and the possible limits met by authors.

Technical comments: Line 21: write "are" instead of "is" in "different cases are considered.." Line 68: add "to" for "in relation to WF reduction" Line 304: write "... the soils are taken from..." deleting is used Figures 7-8: the text on the Y axis is put in the middle of the graph and cannot be read.

---

## Referee Comment (RC3) · Anonymous Referee #3 · 14 Mar 2017

The authors discuss an important topic about a modeling approach rather than expert-based approach to deriving marginal cost curves for irrigated agriculture. Their paper has a lot of data and detailed analysis and the method they offer seems to be relevant and to work. It involves a lot of data and assumptions and would seem to be laborious in any actual application, although one could envision a software package that would make the computations easier, assuming the data could be obtained.

The authors published a paper on the same general topic (footprint of irrigated agriculture) in this journal, and this work would seem to be an extension of it. I have no

detailed comments on the methodology, which seems to be straightforward and mainly to use a software package to simulate a lot of scenarios and then plot the resulting cost curves. I judge the work to be of publication quality.

I think the article merits publication. My suggestion is to add some text to explain how this work can be used. Who will use it and for which decisions? Is it simply a model exercise meant for the research literature or can the work be translated into action programs?

Two small points: Line 168 there is an error, "covered" needs a "d" on the end; line 226 "installation" would be better than the misspelled "instalment."

---

## Author Comment (AC2) · 22 Mar 2017

**Reply to Anonymous Referee #2**
We thank Referee #2 for the comments; below we give the reply to the comments.

**Comment**
The manuscript is very interesting and focuses on a very important topic: WF reduction studied with the application of MCC for analysing the economic side of strategies improvement for water use reduction. The study is well balanced and clearly written, therefore I suggest accepting it after solving few comments.
Specific comments:
**In the introduction literature is lacking, more details should be given on MCC, on possible studies that tried to perform something similar, and better explaining the advantages and innovation of introducing such an assessment (e.g., pay more attention on lines 67-70). In addition, the references reported are written often together (line 64: 4 references in the brackets) and explaining their single specific role as reference would be helpful.**

**Reply:** We agree to the reviewer's comment on the importance of introducing the cost side of analysing WF reduction, and to the comments on explaining the MCC with additional literature. The following explanations will be incorporated in the revised version of the paper.

The MCC for WF reduction ranks WF reduction measures according to their cost-effectiveness (WF reduction achieved per USD) and shows the most cost-effective set of measures for a certain WF reduction target. The MCC has been applied extensively for carbon footprint reduction in various studies, focusing on various sectors and regions. Enkvist et al. (2007) show cost curves for reducing greenhouse gas emissions of different regions globally. Lewis and Gomer (2008) develop an MCC for reducing greenhouse gas emissions of all sectors in Australia, and (MacLeod et al., 2010) develop an MCC for the agricultural sector in the UK. A detailed method to derive MCCs for the most economically efficient reductions in greenhouse gas emissions in the agricultural sector is presented by (Bockel et al., 2012). The weaknesses and strengths intrinsic to different methods of deriving MCCs of greenhouse gas reduction are reviewed in different papers (Kesicki, 2010;Kesicki and Strachan, 2011;Kesicki and Ekins, 2012). The application of MCCs in the water sector is just starting. Tata-Group (2013) developed MCCs for WF reduction in some factories. Addams et al. (2009) apply MCCs for closing the gap between water supply and demand in irrigated agriculture. Khan et al. (2009) discuss two possible pathways to reduce the environmental footprints of water and energy inputs in food production: improving water productivity and energy use efficiency. This work however does not explicitly specify the measures and their cost effectiveness, which would inform the unit cost of improving water and energy use efficiency. In the current study, we apply MCCs for WF reduction and show the cost effectiveness of measures as well as the most cost efficient pathway to reduce water consumption to a certain WF reduction target, e.g. towards a given WF permit or to a certain WF benchmark.

**Comment**

**Lines 208-213: what are the average yields for the crops? Maize is considered cultivated in Italy only, or also in Spain for example? What about the other crops? With this point in mind, are the Figures 4-5 referred to maize production in one single country or in more? Understanding the country would make possible to connect these results with the values reported in the Appendix.**

**Reply:**

In lines 208 -2013 we state that the WF of crop production is expressed in two ways: WF per tonne of crop production (expressed in $m^3$ per tonne of crop), or WF per unit area (expressed in $m^3$ per hectare per season). Maize, tomato and potato are cultivated in Italy, Spain, Israel and UK. In a previous paper we show the water footprint of each crop for each country (Chukalla et al., 2015). Figures 4-5 in the current paper illustrate the average WF ($m^3$ per tonne) for maize over different cases (four countries, three hydrologic years and three soil types). The whiskers around the WF estimates in Figure 4 show the range of outcomes for the different cases. The WF and cost values reported in Appendix G are also average over the cases and the countries respectively. In the current paper, we consider the average WF estimates, which means that the MCCs are not case specific; this does not affect the goal of the study, which is to introduce the methodological development of an MCC and show its application by giving a hypothetical example.

**Comment**

**Line 250-251: not considering energy for transport and pumping is a very important simplification and surely affects the results. Please motivate your choice.**

**Reply:**

We agree with the referee's comment. The energy costs to bring water from a source to the field (and the water losses during transport) are worth to include, but these are case specific and beyond the scope of the current study. We constrained the study to costs at field level. This does not affect the conclusions from the study as we will explain. The overall annual cost per measure increases if we add the cost of energy to bring irrigation water from a source to a field to the annual cost of a measure at field level. The cost increase will depend on the measure (Figure 1): the cost increase is highest for furrow irrigation, followed by sprinkler and drip or subsurface drip irrigation; furthermore, the cost increase is more with full irrigation than with deficit irrigation; and finally, the cost increase is most with no-mulching followed by organic and synthetic mulching). One can see that the additional cost related to energy for transporting the water to the field decreases in the direction of decreasing WF. Thus, this does not affect the order of measures ranked based on their cost effectiveness in reducing WF.

[Figure]

Figure 1: Average WF per area (m³ ha⁻¹) for maize production and average annual costs associated with 20 management packages. The whiskers around WF estimates indicate the range of outcomes for the different cases (different environments, soils and hydrologic years). The whiskers around cost estimates indicate uncertainties in the costs. WF estimates are split up in blue and green components; costs are split up in investment, water, energy and labour costs. The energy cost to bring irrigation water from a source to a field is calculated by multiplying the volume of irrigation water by a cost of energy, assumed 0.2 $ per m³.

**Comment**

\# Finally, a discussion paragraph is missing, which would be helpful for better discussing literature, the benefits of this new method applied to WF and the possible limits met by authors.

**Reply:**

In the revised paper we will add the following discussion points: (a) discussion of the challenge of developing MCCs for each specific case, requiring a modelling effort for each specific case, (b) discussion of uncertainties involved when assessing the effect of certain measures through modelling, and the variability of effects related to climate variability, (c) a reflection on the problems of obtaining case-specific data on costs, (d) the potential to derive more general conclusions on the ranking of specific measures regarding their cost-effectiveness in WF reduction through more research across cases (different crops and different regions), and (e) the future relevance of MCCs for WF reduction once companies start aiming to reduce the WF of their crop products down to sector-agreed regional-specific benchmark levels and once governments are going to apply farm-specific WF permits based on the maximum sustainable WF in a catchment.

**Technical comments:  (all minor comments are incorporated)**
**Line 21: write "are" instead of "is" in "different cases are considered..."**
**Line 68: add "to" for "in relation to WF reduction"**
**Line 304: write "... the soils are taken from..." deleting is used**
**Figures 7-8: the text on the Y axis is put in the middle of the graph and cannot be read.**

References:
Addams, L., Boccaletti, G., Kerlin, M., and Stuchtey, M.: Charting our water future: economic frameworks to inform decision-making, McKinsey & Company, New York, 2009.
Bockel, L., Sutter, P., Touchemoulin, O., and Jönsson, M.: Using marginal abatement cost curves to realize the economic appraisal of climate smart agriculture policy options, Methodology, 3, 2012.
Chukalla, A. D., Krol, M. S., and Hoekstra, A. Y.: Green and blue water footprint reduction in irrigated agriculture: effect of irrigation techniques, irrigation strategies and mulching, Hydrol. Earth Syst. Sci., 19, 4877-4891, 2015.
Enkvist, P., Nauclér, T., and Rosander, J.: A cost curve for greenhouse gas reduction, McKinsey Quarterly, 1, 34, 2007.
Kesicki, F.: Marginal abatement cost curves for policy making–expert-based vs. model-derived curves, Energy Institute, University College London, 2010.
Kesicki, F., and Strachan, N.: Marginal abatement cost (MAC) curves: confronting theory and practice, Environ Sci Policy, 14, 1195-1204, DOI 10.1016/j.envsci.2011.08.004, 2011.
Kesicki, F., and Ekins, P.: Marginal abatement cost curves: a call for caution, Clim Policy, 12, 219-236, Doi 10.1080/14693062.2011.582347, 2012.
Khan, S., Khan, M., Hanjra, M., and Mu, J.: Pathways to reduce the environmental footprints of water and energy inputs in food production, Food Policy, 34, 141-149, 2009.
Lewis, A., and Gomer, S.: An Australian cost curve for greenhouse gas reduction, Report for McKinsey and Company Australia, 2008.
MacLeod, M., Moran, D., Eory, V., Rees, R., Barnes, A., Topp, C. F., Ball, B., Hoad, S., Wall, E., and McVittie, A.: Developing greenhouse gas marginal abatement cost curves for agricultural emissions from crops and soils in the UK, Agr Syst, 103, 198-209, 2010.
Tata-Group: Tata Industrial Water Footprint Assessment: Results and Learning, 2013.

---

## Author Comment (AC3) · 22 Mar 2017

**Reply to Anonymous Referee #3**
We thank Referee #3 for the comments; below we give the reply to the comments.

**Comment**

The authors discuss an important topic about a modelling approach rather than expert based approach to deriving marginal cost curves for irrigated agriculture. Their paper has a lot of data and detailed analysis and the method they offer seems to be relevant and to work. It involves a lot of data and assumptions and would seem to be laborious in any actual application, although one could envision a software package that would make the computations easier, assuming the data could be obtained.
The authors published a paper on the same general topic in this journal, and this work would seem to be an extension of it. I have no detailed comments on the methodology, which seems to be straightforward and mainly to use a software package to simulate a lot of scenarios and then plot the resulting cost curves. I judge the work to be of publication quality.
I think the article merits publication. My suggestion is to add some text to explain how this work can be used. Who will use it and for which decisions? Is it simply a model exercise meant for the research literature or can the work be translated into action programs?

**Reply:** the current study indeed extends our previous work on green and blue WF reduction of irrigated crops (Chukalla et al., 2015). The modelling approach in deriving marginal cost curves relates to the yield response to field management, water stress and local conditions. The consideration of a large number of combined management options led us to use modelling to assess effects of field scale measures and their interaction, as field experiments are limited in covering such combinations; here we draw from Chukalla et al (2015). The calculation of marginal costs of WF reductions is methodologically straightforward. The derivation of plausible WF reduction pathways requires insight in the agronomic plausibility of successive implementation of field scale measures. Next, the derivation of the marginal cost curve for WF reduction again is in itself straightforward but still uncommon in the WF literature.
The previous paper estimated the WF reduction of different measures and combinations of measures. The current paper, through the MCC, shows the cost-effectiveness of measures and combinations of measures. Therefore, the current paper gives important information for decision making on WF reductions by farmers, using both the cost and physical effect to rank measures. As the referee suggests, it is important to show how the work can be used, who can use it, and for which decisions it can be used; this is explained in the paper in section 3.4: the application of the marginal cost curve.
In reaction to a similar comment of referee #2 we will address in the revised paper the relevance of the work for farmers, companies in the food and beverage sector and water managers. The MCCs presented in the paper are of interest to farmers who are seeking to or incentified to reduce crop water footprints in their production practice. They are of interest of companies in the food and beverage sector that increasing formulate water efficiency targets for their supply chain and that are seeking to stimulate

farmers to reduce their WF. Finally, the MCCs are of interest to water managers responsible for water allocation and supply to irrigation farming, providing them with information on the costs to farmers if they reduce WF permits to farmers.

References:

Chukalla, A. D., Krol, M. S., and Hoekstra, A. Y.: Green and blue water footprint reduction in irrigated agriculture: effect of irrigation techniques, irrigation strategies and mulching, Hydrol. Earth Syst. Sci., 19, 4877-4891, 10.5194/hess-19-4877-2015, 2015.

---

## Referee Comment (RC4) · Anonymous Referee #2 · 27 Mar 2017

Thanks to the Authors for the reply. I feel they answered to all points. However, I still have a question. I can understand this is an example of the methodology adopted and that the focus is not specifically on the single crops of the single countries, but maize cannot be cultivated in UK due to local weather and temperatures. Saying that maize is cultivated there is a conceptual mistake and I cannot understand how you could find data to calculate its Water Footprint.

---

## Referee Comment (RC5) · Anonymous Referee #4 · 29 Mar 2017

The paper tries to derive marginal costs curves for water footprint reductions. In summary, the paper calculates costs and savings of standard practices using the standard tool Aquacrop for 3 european locations plus Israel. The concept is straightforward and therefore of limited originality. The author claim to be the first doing this type of analysis, while later in the paper they admit it has been done previously in the same context but slightly different condition. The difference is relation to water footprint, which is in this context very specifically defined (and only late in the paper). The definition deviates from international standards and should be introduced early in the paper (acknowledge the different definitions and provide rationale for the chosen approach). Additionally, the

scope is very narrow (selecting 4 locations) but not having actual case study data and thus being theoretically. However, the topic as such is not novel and might be suitable for an irrigation journal.

All regions are high income countries and therefore this needs to be stressed in the title (add in high income regions). Also cost data is partially only representative of EU conditions.

One major flaw in the analysis is the lack of accounting of important costs such as fertilizer and land, which is completely omitted but highly important, since land is typically limited and therefore deficit irrigation has a yield reduction (which is a land cost increase). The results are therefore to be reconsidered.

L 14 The authors write about water footprint permit per hectare, which needs to have a rationale

Introduction in general: The authors mainly cite their own work, while it is important to give a broader overview, especially in a diverse field such as water footprint, where many different water footprint concepts have been published and the reader needs to be informed what is done here and how this relates to other work.

L100: here it is important to talk about land use costs too

Section 2.2 is basically directly summarizing Aquacrop and might be moved to Appendix as it does not provide important additional information (at least it can be summarized).

L188ff: The authors talk about green water and differentiating it form blue water. However, there is no way this can be properly done nor is there any hydrological definition of green waster vs. blue water. However, if the concept is used, literature refers to green water as soil moisture (see Falkenmark et al) and thus this is in conflict with previous research. I suggest to omit as it does not add meaningfull information.

L204-206: What about fertilizer use? Is this assumed to be optimal? What are the cost

related to it?

L208. The authors define water footprint cutting supply chain (which is comonly icluded even by the author's own references on water footprint). E.g. seedling and fertilizer water use should at least be mentioned.

Section 2.4. Since major aspects (land and fertilizer costs) are omitted the analysis is very theroetical and not covering the full picture. Additionally the author mix data form global sources and European, while it is used for the European context. Adjustments to price levels needs to be discussed. Some data is really old (1992)

L260. Where is the StDev presented?

L293: the scope of the study (locations etx) should be presented at the beginning, since up to this point the reader supposes it is a globally relevant study.

Results: the results must be revised after inclusion of the additional costs. Also they should report uncertainties. this also applies to the discussion. Especially the main finding that deficit irrigation is useful (even win-win based on this study) the authors should also discuss why there is still potential for it and why it has not been done already (what are the constraints etc).

---

## Editor Comment (EC1) · N. Romano (Editor) · 30 Mar 2017

Dear Authors, Your submission has received general positive comments, but also some criticisms that deserve attention and adequate prelim2nary responses. Please, give due account to the additional point from Ref.# . The questions raised by Ref.#4 also require adequate replies. I hope you can do this within the end of the open discussion step, but let me know if you may want to have a small extension of this deadline.

---

## Author Comment (AC4) · 3 Apr 2017

**Reply to Anonymous Referee #2**

We thank Referee #2 for the comments; below we give the reply to the comments.

**Comment**

Thanks to the Authors for the reply. I feel they answered to all points. However, I still have a question. I can understand this is an example of the methodology adopted and that the focus is not specifically on the single crops of the single countries, but maize cannot be cultivated in UK due to local weather and temperatures. Saying that maize is cultivated there is a conceptual mistake and I cannot understand how you could find data to calculate its Water Footprint.

**Reply:**

The referee's comment is correct in that UK grows little grain maize and the majority used in the country is imported (Statistics-GOV.UK, 2017a). However, maize is still cultivated under both rain-fed and irrigation conditions; according to FAO-aquastat (2017) the irrigated maize is sown in April and covering 4,300 ha in 2007. The total maize cultivation in UK covered 197,000 ha in 2016 (Statistics-GOV.UK, 2017b). According to (Statistics-GOV.UK, 2017b, a) most of the maize production in UK is used for fodder, followed by bioenergy and grain maize: 76% fodder maize, 19% bioenergy maize, and 5% grain maize in 2015. Water footprint calculations were done based on simulations with the AquaCrop model using general validated crop files (Steduto et al., 2011) and local specific growing conditions. AquaCrop simulated maize yields in the UK in the range of 9.5 to 13 t ha$^{-1}$ for different soil types and hydrological years, which is in agreement to the maize yield 9 to 11 t ha$^{-1}$ reported in literature (Marsh, 2017).

References:

FAO-aquastat: FAO: On-line database, United Kingdom irrigated crop calendar report. Food and Agricultrue Organization of the United Nations, http://www.fao.org/nr/water/aquastat/countries_regions/GBR/GBR-CC_eng.pdf, last access: March, 2017.

Marsh, S. P.: Crimped maize grain for finishing beef cattle report for 2010. Animal Science Research Center, http://beefandlamb.ahdb.org.uk/wp/wp-content/uploads/2013/04/crimpedmaizegrainproductionstudy_16.09.10-Final-Report-Production-Study.pdf, last access: March, 2017.

Statistics-GOV.UK: On-line database, Area of crops grown for bioenergy in England and the UK: 2008 - 2014. Department for Environment Food & Rural Affairs, https://www.gov.uk/government/uploads/system/uploads/attachment_data/file/483812/nonfood-statsnotice2014-10dec15.pdf, last access: March, 2017a.

Statistics-GOV.UK: On-line database, Farming statistics final crop areas, yields, livestock populations and agricultural workforce at June 2016. Department for Environmen Food & Rural Affairs, https://www.gov.uk/government/uploads/system/uploads/attachment_data/file/579402/structure-jun2016final-uk-20dec16.pdf, last access: March, 2017b.

Steduto, P., Hsiao, T., Raes, D., Fereres, E., Izzi, G., Heng, L., and Hoogeveen, J.: Performance review of AquaCrop– The FAO crop-water productivity model, ICID 21st International Congress on Irrigation and Drainage, 2011, 15-23,

---

## Author Comment (AC5) · 10 Apr 2017

**Reply to Anonymous Referee #4**
We thank Referee #4 for the comments; below we give the reply to the comments.

**Comment**
The paper tries to derive marginal costs curves for water footprint reductions. In summary, the paper calculates costs and savings of standard practices using the standard tool Aquacrop for 3 european locations plus Israel. The concept is straightforward and therefore of limited originality. The author claim to be the first doing this type of analysis, while later in the paper they admit it has been done previously in the same context but slightly different condition. The difference is relation to water footprint, which is in this context very specifically defined (and only late in the paper). The definition deviates from international standards and should be introduced early in the paper (acknowledge the different definitions and provide rationale for the chosen approach). Additionally, the scope is very narrow (selecting 4 locations) but not having actual case study data and thus being theoretically. However, the topic as such is not novel and might be suitable for an irrigation journal.

**Reply:**
We agree with the referee comment that the definition of water footprint (WF) is introduced later in the paper. In the introduction part of the revised paper, we will add the WF definition and the rationale on the chosen approach: "*Reducing the non-beneficial consumptive water loss is a means of demand management to increase water use efficiency and thus reduce water scarcity (Mekonnen and Hoekstra, 2016). In crop production, reducing the non-beneficial consumptive water use is possible by decreasing the ratio of evapotranspiration (ET) over the growing period to the crop yield (Y) or increasing the inverse ratio (Y to ET). The first ratio is called the water footprint of crop production (Hoekstra et al., 2011); the inverse ratio is called water productivity (Molden et al., 2010).*"

There are model-driven and expert-based approaches to develop the MCC; the two approaches have been applied extensively in *carbon footprint* reduction. In the area of WF reduction, the expert-based approach has been applied only once, in a case for the industrial sector (Tata-Group, 2013). The current paper pioneers by introducing a *model-driven MCC* in the area of WF reduction and by applying a MCC for water footprint reduction in *irrigated agriculture*. In addition to MCCs, we show the plausible WF reduction pathways, which requires insight in the agronomic plausibility of successive implementation of field scale measures. The approach is highly original and much needed, filling the gap of existing literature on water footprint reduction which generally lacks the practical and economic component: what are the subsequent steps and associated costs to achieve increasing levels of water footprint reduction.

The referee is right we did not use field measured data for validating the simulated result. This limitation puts a disclaimer to the simulated results of our study, but we believe that the results of this study provide a useful reference for similar future studies with other models. Basically we develop an advanced method to develop MCCs and WF reduction pathways for irrigated agriculture.

**Comment**

**All regions are high income countries and therefore this needs to be stressed in the title (add in high income regions). Also cost data is partially only representative of EU conditions.**

**Reply:**

We will add the description of the case study areas in the abstract and introduction section of the revised version of the paper.

**Comment**

**One major flaw in the analysis is the lack of accounting of important costs such as fertilizer and land, which is completely omitted but highly important, since land is typically limited and therefore deficit irrigation has a yield reduction (which is a land cost increase). The results are therefore to be reconsidered.**

**Reply:**

The study is not a full cost benefit analysis of crop production. The study is a cost-effectiveness study to achieve a certain water footprint reduction. We include marginal costs and benefits associated with changing different components in the overall management practice. Costs of fertilizers are a highly relevant element of cost for a farmer, but we don't change fertilizer application practice, so there are no changes in this respect. The costs of land are also relevant, but we consider a fixed field, which means no change in land costs. Moving from full irrigation to deficit irrigation indeed involves yield reduction, but in our study yield reduction from deficit irrigation is kept under 2%.

**Comment**

**L 14 The authors write about water footprint permit per hectare, which needs to have a rationale**

**Reply:**

With the overall increasing demand for water and effect of climate on water availability (Hanjra and Qureshi, 2010), there is fierce competition among various water using sectors. The rationale behind a WF permit per hectare is that the water availability per catchment is limited to runoff minus environmental flow requirement. When dividing the maximum amount of water available in a catchment over the croplands that need irrigation, one finds a maximum volume of water available per hectare of cropland. This could be translated in water allocation policy into a max WF permit per hectare. This is just one way of promoting water footprint reduction in areas where that is needed. Another way is to create incentives to reduce the WF per unit of production to a certain benchmark level. The MCCs we develop can be used for analysing cost-effective WF reduction given either a target level for WF per unit of crop or a target level for WF per hectare.

**Comment**

**Introduction in general: The authors mainly cite their own work, while it is important to give a broader overview, especially in a diverse field such as water footprint, where many different water footprint concepts have been published and the reader needs to be informed what is done here and how this relates to other work.**

**Reply:**

We are aware as there are debates on water footprint concepts and methodological assessment phases between the life cycle assessment (LCA) and water footprint assessment (WFA) approaches (Chenoweth et al., 2014;Pfister et al., 2017;Hoekstra, 2016). The two approaches differ in their impact/sustainability assessment stage, not in the way they account water consumption in their inventory/accounting stage (Boulay et al., 2013). The current study focuses on cost-effective reduction of water consumption from a field, whereby it does not make a difference on whether we take a WFA or LCA perspective.

**Comment**

**L100: here it is important to talk about land use costs too**

**Reply:**

The study is on MCC for reducing WF per hectare, which informs us about cost of saving water per ha, and on MCC for reducing WF per tonne, which informs us about cost of increasing water use efficiency. We do not aim to represent an agro-economic cost-benefit analysis with inclusion of the costs of all input factors and all revenues. We make sure that yields remain within 2% of the max yield, so that is why we left out reduced revenue at the cost side. We will better highlight this in the paper (now this remark is hidden in method section 2.2).

**Comment**

**Section 2.2 is basically directly summarizing Aquacrop and might be moved to Appendix as it does not provide important additional information (at least it can be summarized).**

**Reply:**

In Section 2.2 we don't really summarize Aquacrop, but highlight the way we implement the different measures in Aquacrop, just to be fully transparent about our analysis.

**Comment**

**L188ff: The authors talk about green water and differentiating it form blue water. However, there is no way this can be properly done nor is there any hydrological definition of green waster vs. blue water. However, if the concept is used, literature refers to green water as soil moisture (see Falkenmark et al) and thus this is in conflict with previous research. I suggest to omit as it does not add meaningful information.**

**Reply:**

We follow the broadly agreed interpretation that blue water in the soil refers to irrigation water (derived from groundwater or surface water) and that green water in the soil stems from rainwater (Falkenmark and Rockström, 2006). The green and blue fractions in total ET are calculated based on the green to blue water ratio in the soil moisture, which in turn is kept track of over time by accounting for how much green

and blue water enter the soil moisture, following the accounting procedure as reported in Chukalla et al. (2015).

**Comment**
**L204-206: What about fertilizer use? Is this assumed to be optimal? What are the cost related to it?**
**Reply:**
Yes, we assumed optimal fertilizer. The cost of fertilizer is not considered as it is not required to develop the objective of the paper, developing MCCs for reducing WF.

**Comment**
**L208. The authors define water footprint cutting supply chain (which is commonly included even by the author's own references on water footprint). E.g. seedling and fertilizer water use should at least be mentioned.**
**Reply:**
The WF of a *product* is equal to the WF of the *processes* to produce the product, considering all processes over the supply chain (Hoekstra et al., 2011). In the current study we just focus on the process of crop growing, hence we just look at the WF at field level, not the WF of inputs to the process. We will highlight that in the paper.

**Comment**
**Section 2.4. Since major aspects (land and fertilizer costs) are omitted the analysis is very theroetical and not covering the full picture. Additionally the author mix data form global sources and European, while it is used for the European context. Adjustments to price levels needs to be discussed. Some data is really old (1992)**
**Reply:**
Using up-to-date cost data that reflect the market value of the technologies would have been our interest, but we notice that getting such data is difficult. For our purpose, showing a method rather than evaluate a specific case, the data we used suffice. We will advise readers to use the paper as a guide to develop MCCs for WF reduction rather than take all data for granted. Each specific case will require its own data.

**Comment**
**L260. Where is the StDev presented?**
**Reply:**
We indicate the uncertainty around the cost using whiskers in Fig. 3, 4, 7 and 8.

**Comment**
**L293: the scope of the study (locations etx) should be presented at the beginning,**
since up to this point the reader supposes it is a globally relevant study.
**Reply:**
We will add the location of the study both in the abstract and introduction section of the revised paper.

**Comment**

**Results: the results must be revised after inclusion of the additional costs. Also they should report uncertainties. this also applies to the discussion. **Especially the main finding that deficit irrigation is useful (even win-win based on this study) the authors should also discuss why there is still potential for it and why it has not been done already (what are the constraints etc).**

**Reply:**

We will reflect on the cost uncertainties and why deficit is not practiced yet in the discussion and conclusion section of the revised version of the paper.

References:

Boulay, A.-M., Hoekstra, A. Y., and Vionnet, S.: Complementarities of water-focused life cycle assessment and water footprint assessment, Environ. Sci. Technol, 47, 11926-11927, 2013.

Chenoweth, J., Hadjikakou, M., and Zoumides, C.: Quantifying the human impact on water resources: a critical review of the water footprint concept, Hydrol Earth Syst Sc, 18, 2325-2342, 2014.

Falkenmark, M., and Rockström, J.: The new blue and green water paradigm: breaking new ground for water resources planning and management, Journal of Water Resources Planning and Management, 132, 129-132, 2006.

Gómez-Limón, J. A., and Martinez, Y.: Multi-criteria modelling of irrigation water market at basin level: A Spanish case study, European journal of operational research, 173, 313-336, 2006.

Hanjra, M. A., and Qureshi, M. E.: Global water crisis and future food security in an era of climate change, Food Policy, 35, 365-377, 2010.

Hoekstra, A. Y., Chapagain, A. K., Aldaya, M. M., and Mekonnen, M. M.: The Water Footprint Assessment Manual: Setting the Global Standard, Earthscan, London, UK, 2011.

Hoekstra, A. Y.: A critique on the water-scarcity weighted water footprint in LCA, Ecol Indic, 66, 564-573, 2016.

Mekonnen, M. M., and Hoekstra, A. Y.: Four billion people facing severe water scarcity, Science advances, 2, e1500323, 2016.

Molden, D., Oweis, T., Steduto, P., Bindraban, P., Hanjra, M. A., and Kijne, J.: Improving agricultural water productivity: between optimism and caution, Agricultural Water Management, 97, 528-535, 2010.

Pfister, S., Boulay, A.-M., Berger, M., Hadjikakou, M., Motoshita, M., Hess, T., Ridoutt, B., Weinzettel, J., Scherer, L., and Döll, P.: Understanding the LCA and ISO water footprint: A response to Hoekstra (2016)"A critique on the water-scarcity weighted water footprint in LCA", Ecol Indic, 72, 352-359, 2017.

Tata-Group: Tata Industrial Water Footprint Assessment: Results and Learning, 2013.

---

## Author Comment (AC6) · 10 Apr 2017

Dear Prof. Nunzio Romano, We thank you for the suggested extension on the deadline; we finally managed posting the reply to referee #4 within the time open for discussion.

Kind regards, Abebe
* * *